# Molecular identity of proprioceptor subtypes innervating different muscle groups in mice

Stephan Dietrich [1], Carlos Company[2], Kun Song [3],
Elijah David Lowenstein [4,5], Levin Riedel [1], Carmen Birchmeier [4,5],
Gaetano Gargiulo [2] & Niccolò Zampieri [1] ✉

The precise execution of coordinated movements depends on proprioception, the sense of body position in space. However, the molecular underpinnings of proprioceptive neuron subtype identities are not fully understood. Here we used a single-cell transcriptomic approach to define mouse proprioceptor subtypes according to the identity of the muscle they innervate. We identified and validated molecular signatures associated with proprioceptors innervating back (*Tox*, *Epha3*), abdominal (*C1ql2*), and hindlimb (*Gabrg1*, *Efna5*) muscles. We also found that proprioceptor muscle identity precedes acquisition of receptor character and comprise programs controlling wiring specificity. These findings indicate that muscle-type identity is a fundamental aspect of proprioceptor subtype differentiation that is acquired during early development and includes molecular programs involved in the control of muscle target specificity.

Proprioception, the sense of body position in space, is critical for the generation of coordinated movements and reflexive actions. The primary source of proprioceptive information is represented by sensory neurons in the dorsal root ganglia (DRG), whose afferents innervate specialized mechanoreceptive organs detecting muscle stretch and tension[1]. Proprioceptive sensory neurons can be anatomically and functionally divided on the basis of the identity of the muscle and receptor organ they innervate. First, during early development, proprioceptors innervate muscles and in order to precisely adjust motor output according to the biomechanical properties of their targets wire with neural circuits in the central nervous system (CNS) with exquisite specificity[2,3]. In addition, at a receptor level, proprioceptors can be further distinguished into three subtypes - Ia, Ib, and II - by their selective contribution to either muscle spindles (MS; Ia and II) or Golgi tendon organs (GTO; Ib)[4]. Most notably, Ia sensory afferents make monosynaptic connections to motor neurons controlling the activity of the same muscle, as well as synergist muscle groups, while avoiding motor neurons controlling the function of antagonist muscles, thus providing the anatomical substrate for the stretch reflex[5,6]. These precise patterns of connectivity are conserved in all limbed vertebrates and their assembly precedes the emergence of neural activity[7,8], implying that proprioceptive neurons are endowed from early developmental stages with molecular programs controlling critical features of their muscle-type identity, such as central and peripheral target specificity[9,10]. However, these determinants are still largely unknown, thus hindering efforts to define the mechanisms underlying the development of proprioceptive sensory neuron subtypes, the wiring of spinal sensorimotor circuits, and the contribution of muscle-specific proprioceptive feedback to motor control.

[1]Laboratory of Development and Function of Neural Circuits, Max-Delbrück-Center for Molecular Medicine, Robert-Rössle-Str. 10, 13125 Berlin, Germany. [2]Laboratory of Molecular Oncology, Max-Delbrück-Center for Molecular Medicine, Robert-Rössle-Str. 10, 13125 Berlin, Germany. [3]Brain Research Center and Department of Biology, School of Life Sciences, Southern University of Science and Technology, Shenzhen 518055 Guangdong, China. [4]Laboratory of Developmental Biology/Signal Transduction, Max-Delbrück-Center for Molecular Medicine, Robert-Rössle-Str. 10, 13125 Berlin, Germany. [5]Neurowissenschaftliches Forschungzentrum, NeuroCure Cluster of Excellence, Charité; Charitéplatz 1, 10117 Berlin, Germany. ✉e-mail: niccolo.zampieri@mdc-berlin.de

Single-cell transcriptomic efforts have revealed remarkable diversity among the major types of somatosensory neurons, while proprioceptors, despite their evident functional heterogeneity, seemed to represent a relatively more homogenous population[11–13]. Recent studies aimed at characterizing the molecular nature of group Ia, Ib, and II neurons have revealed that signatures for receptor sub-types emerge late during development and are consolidated at post-natal stages[14,15]. However, the molecular basis of proprioceptor muscle-type identity remains elusive and so far only few markers for neurons innervating muscles in the distal hindlimb compartment have been identified[9].

In this study, we used a single-cell transcriptomic approach that takes advantage of the somatotopic organization of proprioceptor muscle innervation to reveal the molecular profiles of cardinal muscle identities - epaxial and hypaxial - defined by peripheral connectivity to back and abdominal muscle groups at thoracic level, and lower back and hindlimb muscles at lumbar level. Our data show that muscle-type identity is acquired and consolidated during embryonic development and precedes the emergence of receptor character. In addition, we found that the identified molecular signatures comprise programs controlling defining features of proprioceptor muscle character, such as the specificity of muscle connectivity. In particular, differential expression of axon guidance molecules of the ephrin-A/EphA family discriminates epaxial and hypaxial muscle identities and elimination of ephrin-A5 function erodes the specificity of peripheral connectivity. Altogether, this study reveals that muscle-type identity is a fundamental aspect of proprioceptor subtype differentiation that is acquired during early development and includes molecular programs involved in the control of muscle target specificity.

## Results

### Transcriptome analysis at e15.5 reveals distinct proprioceptive clusters

In order to identify molecular correlates of proprioceptive sensory neurons (pSN) muscle identity we used transcriptome analysis of neurons isolated from thoracic and lumbar DRG at embryonic day (e) 15.5. At this stage proprioceptors have just reached muscle targets in the periphery and their central afferents are progressing toward synaptic partners in the ventral spinal cord (Supplementary Fig. 1a)[4]. In addition, neurons collected from different segmental levels are predicted to reveal traits of epaxial and hypaxial pSN muscle identities, as the cell bodies of neurons innervating back and abdominal muscle groups are found in thoracic DRG, while the ones innervating lower back and hindlimb muscle groups in lumbar DRG (Fig. 1a).

We took advantage of parvalbumin expression in proprioceptors and a small subset of cutaneous mechanoreceptors[16] to isolate 960 neurons - 480 from thoracic (T) levels 1–12 and 480 from lumbar (L) levels 1 to 5 - via fluorescence-activated cell sorting after dissociation of DRG from a BAC mouse line expressing tdTomato under the control of the parvalbumin promoter ($Pv^{tdTom}$)[17] and processed them using the CEL-Seq2 protocol (Supplementary Fig. 1b, c)[18]. 519 neurons passed quality controls (see "Methods" for details) and were found distributed into five molecularly distinct clusters (Fig. 1b and Supplementary Fig. 1d–f). Transcriptome analysis indicated that neurons in cluster 1 represent proprioceptors, as they express general markers of proprioceptive identity ($Pv$, $Runx3$, $Etv1$, and $Ntrk3$). Neurons found in clusters 2–4 present a signature consistent with mechanoreceptor identity ($Maf^+$ and $Ntrk2^+$), with cluster 3 consisting of a postmitotic subset ($Isl1^+$ and $Avil^+$) while clusters 2 and 4 are characterized by proliferation markers ($Mki67^+$, $Mcm2^+$, and $Pcna^+$). Finally, cluster 5 represents neurons contaminated with glial transcripts ($Mpz^+$ and $Apoe^+$; Fig. 1c and Supplementary Fig. 1g)[19].

Next, to highlight differences between proprioceptors in cluster 1 we re-clustered these cells and obtained seven subsets (pS1-pS7; Fig. 1d and Supplementary Fig. 1h). In order to test whether anatomical

provenance could point to proprioceptor muscle-type identities, we assigned segmental origin to each cell. We found that neurons in pS2, pS4, pS5, and pS7 mainly originated from lumbar DRG and therefore could represent proprioceptors connected to hindlimb muscles or the small subset of back muscles found at lumbar levels (lower back and tail muscles), while pS1, pS3, and pS6 mainly arose from thoracic levels, where proprioceptors innervating back and abdominal muscles are located (Fig. 1a, e, f)[20]. We confirmed thoracic and lumbar origin at a transcriptional level by evaluating expression of $Hoxc10$, a gene defining lumbar identity[21], and found that it closely recapitulated the anatomical assignment (Supplementary Fig. 1i). Next, we performed differential gene expression analysis and revealed distinct molecular signatures for each of these clusters (Fig. 1g). Surprisingly, we found that $Trpv1$ is selectively enriched in neurons found in pS6 (Supplementary Fig. 1j). Trpv1 is a well-known marker of nociceptive/thermosensitive neurons and therefore is not expected to be expressed in proprioceptors[22]. Nevertheless, we confirmed the presence of $Pv^+$; $Runx3^+$; $Trpv1^+$ DRG sensory neurons in e15.5 embryos, representing at this stage ~5% of all proprioceptors, both at thoracic and lumbar levels (Fig. 1h, i).

### Embryonic expression of Trpv1 defines a subset of proprioceptors connected to back muscles

Next, to verify whether $Trpv1$ expression in embryonic proprioceptors marks a discrete neuronal subtype we took advantage of mouse lines driving expression of Cre and Flp recombinases under control of the Trpv1 ($Trpv1^{Cre}$)[23] and parvalbumin ($Pv^{Flp}$)[24] promoters to label neurons with an intersectional tdTomato reporter allele ($Trpv1$; $Pv$; $tdT$, $Ai65$)[25].

Anatomical analysis of postnatal day (p) 7 spinal cords, DRG, and muscles from $Trpv1$; $Pv$; $tdT$ mice revealed a well-defined subset of sensory neurons. In the spinal cord, we found labeling of central afferents targeting and making vGluT1+ synaptic contacts with ChAT+ motor neurons in the median motor column (MMC), both at thoracic and lumbar levels, which are known to selectively innervate back and lower back/tail muscles (Fig. 2a, b, Supplementary Fig. 2b, Supplementary Movie 1 and 2). In contrast, limb-projecting motor neurons in the lateral motor column (LMC) received little, if any, input from $Trpv1$; $Pv$; $tdT$ axons (Figs. 2a, b, Supplementary Movie 1)[26]. In agreement with selective central innervation of neurons in the MMC, which is the only motor neuron column present at all rostro-caudal spinal levels, we observed labeling of a subset of parvalbumin+ neurons in cervical, thoracic, and lumbar DRG (Fig. 2c–e and Supplementary Fig. 2c). In the periphery, we found labeling of type Ia, Ib, and II receptors in back but not abdominal muscles (Fig. 2f, Supplementary Fig. 2a, d). Finally, in order to test the overall specificity of lineage tracing in $Trpv1$; $Pv$; $tdT$ mice, we analyzed reporter expression in the brain. We did not find any labeling aside from axons projecting to the dorsal column nuclei of the brainstem that are known to receive direct innervation from proprioceptive sensory neurons (Supplementary Fig. 2e).

In addition, we assessed whether lineage tracing from the Trpv1 promoter ($Trpv1$; $tdT$. $Ai14$)[23,24] would also capture the same population of proprioceptors. Indeed, we observed labeling of a subset of $Pv^+$ neurons in cervical, thoracic, and lumbar DRG, whose central afferents selectively targeted MMC neurons at all segmental levels (Supplementary Fig. 2f, g). Altogether these data show that Trpv1 expression in embryonic proprioceptors defines a subset of proprioceptive sensory neurons selectively innervating back muscles.

### Molecular signatures of proprioceptor muscle-type identities

The opportunity to genetically access a defined subset of proprioceptors defined by their connectivity to the back muscle compartment prompted us to further investigate the molecular identity of back- (Ba-pSN), abdominal- (Ab-pSN), and hindlimb-innervating (Li-pSN) neurons. To this end, we dissociated DRG and manually picked 576 tdTomato+ neurons from thoracic and lumbar levels of $Pv$; $tdT$ ($Pv^{Cre}$;

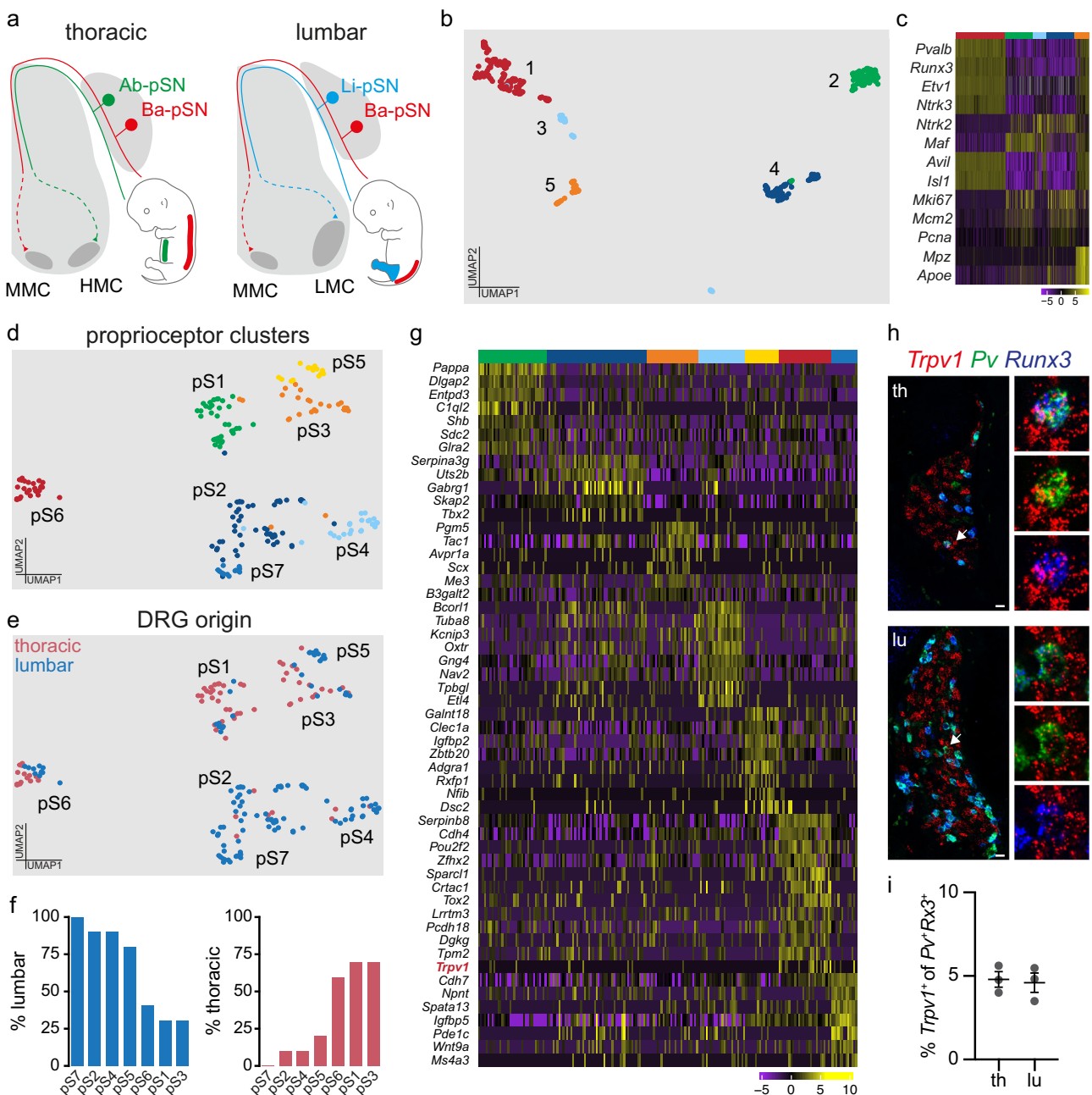

**Fig. 1 | Single-cell transcriptome analysis of thoracic and lumbar proprioceptors at e15.5. a** Schematic illustrating central and peripheral connectivity of e15.5 proprioceptors at thoracic (left) and lumbar (right) spinal levels. Ab-pSN, abdominal muscles-connecting proprioceptors; Ba-pSN, back muscles-connecting proprioceptors; Li-pSN, hindlimb muscles-connecting proprioceptors; MMC, median motor column; HMC, hypaxial motor column; LMC, lateral motor column. **b** UMAP visualization of tdTomato[+] neuron clusters from $Pv^{tdTom}$ embryos at e15.5. **c** Gene expression analysis (logcounts) of proprioceptors (*Pv, Runx3, Etv1, Ntrk3*), mechanoreceptors (*Ntrk2, Maf*), postmitotic neurons (*Avil, Isl1*), proliferating neurons (*Mki67, Mcm2, Pcna*) and glial (*Mpz, Apoe*) markers. **d** UMAP visualization of proprioceptor clusters identified from analysis of cluster 1. **e** UMAP visualization of proprioceptor clusters color-coded according to the thoracic (red) and lumbar (blue) origin of the cells. **f** Percentage of proprioceptors originating from lumbar (left) and thoracic (right) DRG in different proprioceptor clusters. **g** Differential gene expression analysis (logcounts) in proprioceptive clusters (pS1, green; pS2, dark blue; pS3, orange; pS4, light blue; pS5, yellow; pS6, red; pS7, blue).
**h** Representative single molecule fluorescent in situ hybridization (smFISH) images of thoracic (top) and lumbar (bottom) e15.5 DRG sections showing proprioceptors (*Runx3*[+]; *Pv*[+]) expressing *Trpv1*. Scale bar: 25 μm. **i** Percentage of proprioceptors (*Runx3*[+]; *Pv*[+]) expressing *Trpv1* in thoracic and lumbar DRG at e15.5 (mean ± SEM, *n* = 3 animals). Source data are provided as a Source Data file.

*Ai14*; 96 thoracic and 96 lumbar neurons)[27], and *Trpv1; Pv; tdT* mice (192 thoracic and 192 lumbar neurons)[22,23] at p1 and performed single-cell transcriptome analysis (Fig. 3a and Supplementary Fig. 3a). 244 cells passed quality control criteria (see methods for details) and were found distributed into four clusters (C1-C4) expressing high levels of general proprioceptive markers (Fig. 3b and Supplementary Fig. 3b–e). Cluster C1 presented signs of glia contamination and was excluded

from subsequent analysis (Supplementary Fig. 3f). For the remaining clusters, we used mouse line and segmental level of origin of each neuron as means to assign a presumptive muscle-type identity. We found that the majority of cells picked from *Trpv1; Pv; tdT*, thus bona fide Ba-pSN, were found in C2 (Fig. 3c, d and Supplementary Fig. 3g). The majority of lumbar neurons from *Pv; tdT* mice, putative Li-pSN, was found in C3 and the remaining thoracic neurons, by

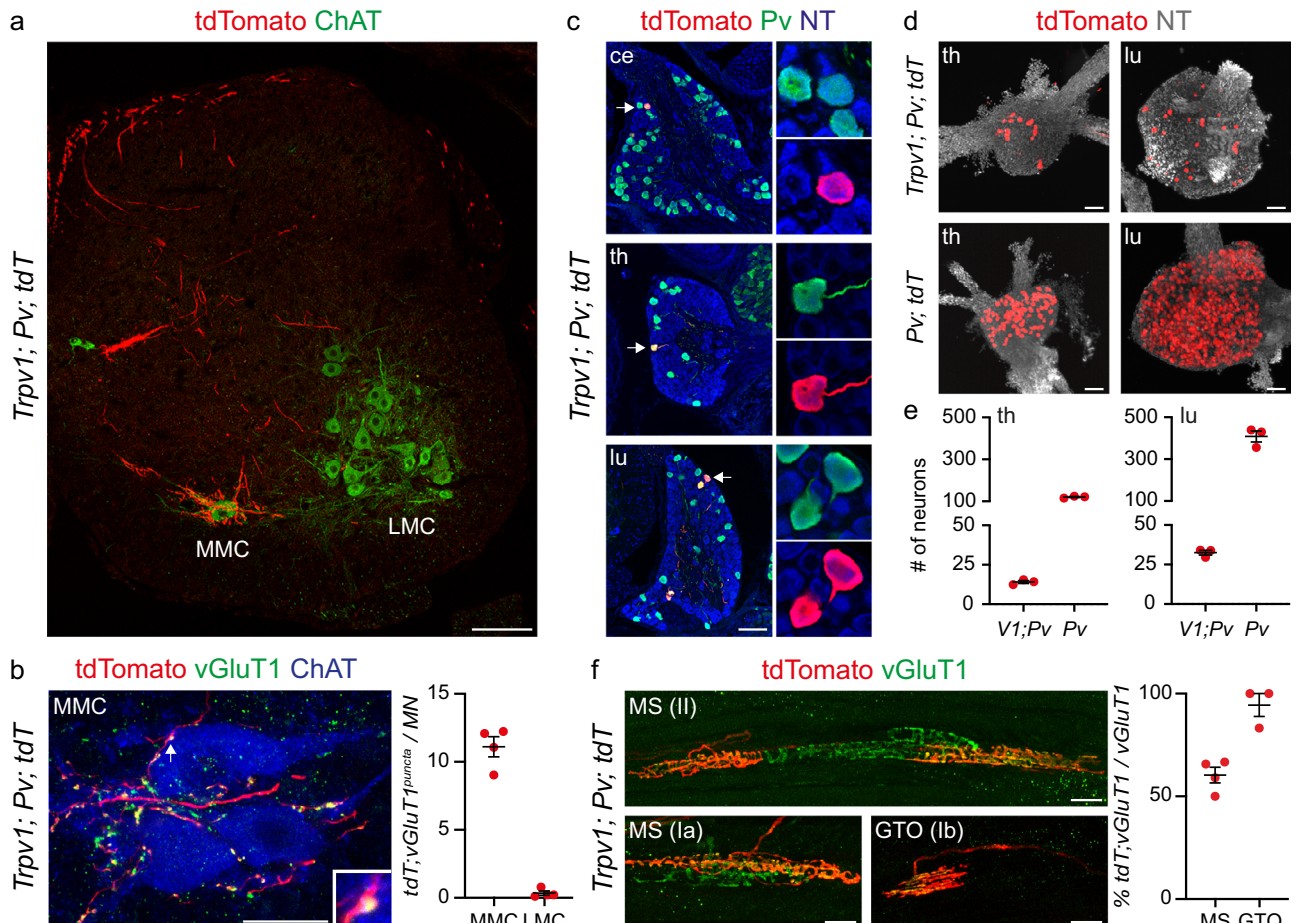

**Fig. 2 | Genetic targeting of proprioceptors innervating back muscles.**
**a** Representative image of tdTomato⁺ afferents in a lumbar spinal cord section from p7 *Trpv1^{Cre}; Pv^{Flp}; Ai65* mice. MMC, median motor column; LMC, lateral motor column. Scale bar: 100 μm. **b** Representative image (MMC, left) and quantification (right) of tdTomato⁺; vGluT1⁺ presynaptic puncta juxtaposed to MMC or LMC neurons cell bodies and proximal dendrites of p7 *Trpv1^{Cre}; Pv^{Flp}; Ai65* mice (mean ± SEM, *n* = 4 animals). Scale bar: 25 μm. **c** Representative images of cervical, thoracic, and lumbar DRG sections showing tdTomato⁺; Pv⁺ neurons in p7 *Trpv1^{Cre}; Pv^{Flp}; Ai65* mice. Scale bar: 100 μm. **d** Whole mount preparations of thoracic (left)

and lumbar (right) DRG showing genetically labeled neurons from p1 *Trpv1^{Cre}; Pv^{Flp}; Ai65* (top) and *Pv^{Cre}; Ai14* (bottom) mice. Scale bar: 100 μm. **e** Number of tdTomato⁺ sensory neurons in DRG from p1 *Trpv1^{Cre}; Pv^{Flp}; Ai65* and *Pv^{Cre}; Ai14* at thoracic (T1-T12, left) and lumbar (L1-L5, right) levels (mean ± SEM, n = 3 animals).
**f** Representative images (left) and quantification (right) of tdTomato⁺ group Ia, II, and Ib afferents in muscle spindles (MS) and Golgi tendon organs (GTO) from the erector spinae muscle of *Trpv1^{Cre}; Pv^{Flp}; Ai65* mice. (mean ± SEM, *n* = 4 animals). Scale bar: 25 μm. Source data are provided as a Source Data file.

exclusions putative Ab-pSN, in C4 (Fig. 3c, d, and Supplementary Fig. 3g). In addition, lumbar origin of each neuron was independently confirmed by analysis *Hoxc10* expression (Supplementary Fig. 3h).

Differential gene expression analysis revealed molecular signatures for presumptive Ba-pSN, Ab-pSN, and Li-pSN (Fig. 3e). To validate these findings, we first analyzed the expression of the top differentially expressed genes, *Tox* (C2, "Ba-pSN"), *Gabrg1* (C3, "Li-pSN"), and *C1ql2* (C4, "Ab-pSN") (Supplementary Fig. 3i), in back-innervating proprioceptors labeled in *Trpv1; Pv; tdT* mice. In agreement with the predicted identity, *Tox* expression was found in nearly all tdTomato⁺ neurons at thoracic and lumbar levels, while *Gabrg1* and *C1ql2* were not (Fig. 3f). Second, we examined expression and DRG distribution in the overall proprioceptive population labeled in *Pv; tdT* mice. At thoracic levels, where proprioceptors innervating back and abdominal muscle groups are located, we observed *Tox* expressed in ~60% of tdTomato⁺ neurons and *C1ql2* in ~28%. At lumbar 3 and 4 levels, where limb-innervating proprioceptors are predominant, we found *Gabrg1* in ~46% of tdTomato⁺ neurons and *Tox* in ~10% (Fig. 3g). Altogether these data indicate that *Tox* and *C1ql2* are expressed within thoracic DRG and *Gabrg1* in lumbar DRG with frequencies expected for Ba-pSN, Ab-pSN and Li-pSN markers (Fig. 3h). Moreover, we found that expression of either *Gabrg1* or *Efna5*, transcripts characterizing cluster

C3, covers ~75% tdTomato⁺ neurons at lumbar level, thus indicating that combination of multiple genes is necessary to define the hindlimb compartment (Supplementary Fig. 3j). Finally, in order to check whether effects of lineage tracing in *Pv; tdT* mice might have influenced these results, we analyzed expression of *Tox, Gabrg1*, and *C1ql2* in *Pv⁺* DRG neurons from wild-type mice and observed similar patterns and frequencies of expression at thoracic and lumbar levels (Supplementary Fig. 3k). Altogether, these data confirm that molecular markers of putative proprioceptor muscle subtypes identified with transcriptome analysis at p1 are expressed in thoracic and lumbar proprioceptive neurons from *Trpv1; Pv; tdT, Pv; tdT*, and wild-type mice with specificity and frequency consistent with back, abdominal and hindlimb muscle identities.

**Proprioceptor muscle identity emerges during early development**
In order to further validate these observations and directly link molecular identity to muscle identity, we investigated expression of markers in proprioceptor subtypes identified by their muscle connectivity. To this end, we examined *Tox* (C2, "Ba-pSN"), *Gabrg1* (C3, "Li-pSN"), and *C1ql2* (C4, "Ab-pSN") expression in DRG neurons retro-gradely labeled after cholera toxin B (CTB) injection in representative

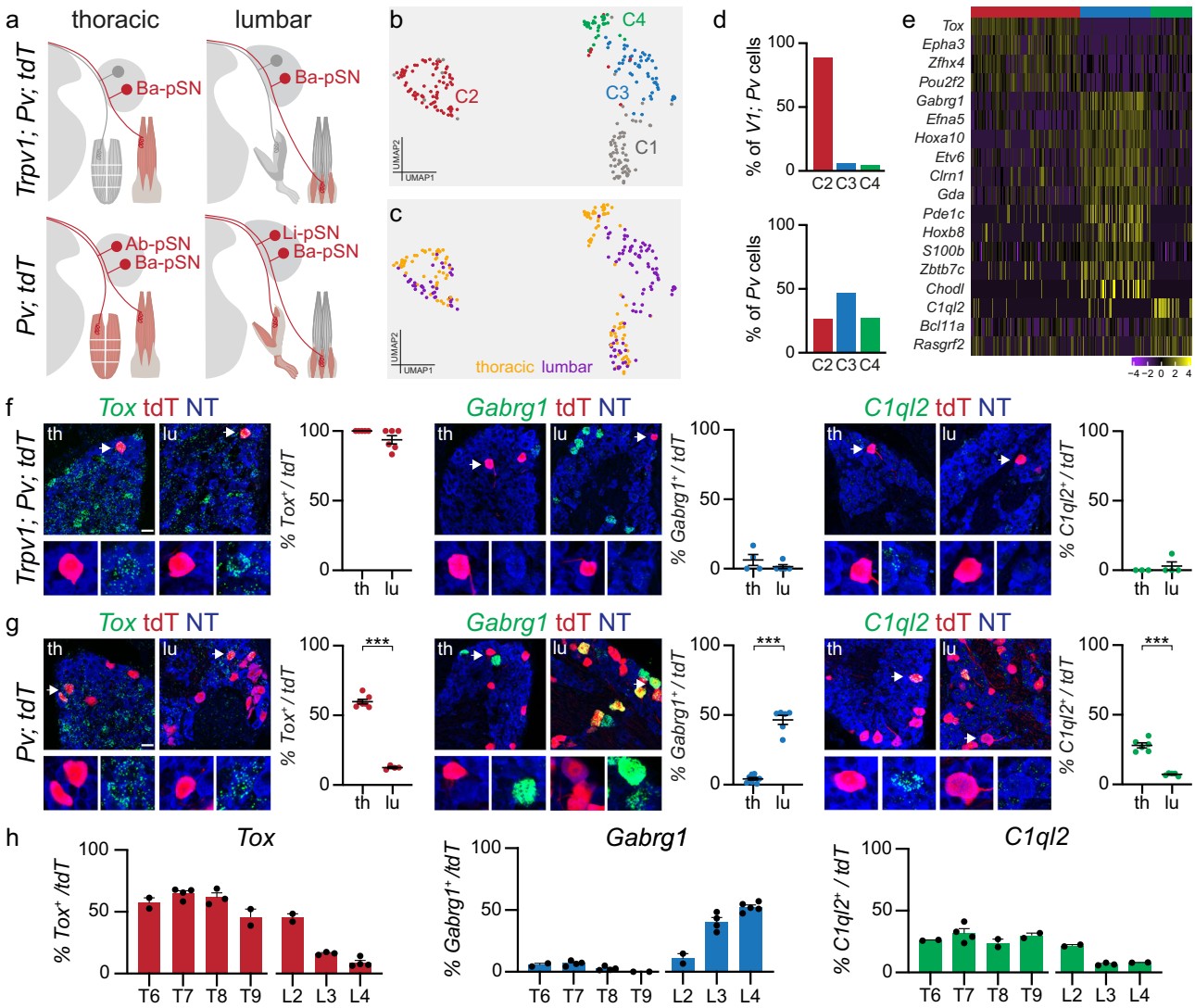

**Fig. 3 | Molecular profiles of back-, abdominal- and hindlimb-innervating proprioceptors. a** Schematics illustrating labeling of the Ba-pSN subset captured in *Trpv1^Cre*; *Pv^Flp*; *Ai65* and all proprioceptors in *Pv^Cre*; *Ai14* at thoracic and lumbar levels. **b** UMAP visualization of cell clusters after transcriptome analysis of tdTomato⁺ DRG neurons from p1 *Trpv1^Cre*; *Pv^Flp*; *Ai65* and *Pv^Cre*; *Ai14* mice at thoracic and lumbar levels. **c** UMAP visualization of cell clusters color-coded according to the anatomical origin (thoracic, yellow; lumbar, purple) of neurons. **d** Bar graph illustrating the percentage of *Trpv1^Cre*; *Pv^Flp*; *Ai65* (top) and *Pv^Cre*; *Ai14* (bottom) cells found in clusters C2 (red), C3 (blue), C4 (green). **e** Differential gene expression analysis (logcounts) for clusters C2 (red), C3 (blue), and C4 (green).

**f** Representative smFISH images and quantification of *Tox* (C2), *Gabrg1* (C3), and *C1ql2* (C4) expression in tdTomato⁺ thoracic and lumbar DRG neurons from p1 *Trpv1^Cre*; *Pv^Flp*; *Ai65* mice (mean ± SEM, n ≥ 3 animals). Scale bar: 25 μm. **g** Representative smFISH images and quantification of *Tox* (C2), *Gabrg1* (C3), and *C1ql2* (C4) expression in tdTomato⁺ thoracic (T6–T9) and lumbar (L3–L4) DRG neurons from p1 *Pv^Cre*; *Ai14* mice (mean ± SEM, n ≥ 4 animals, two-tailed, *t*-test, ***p < 0.001). Scale bar: 25 μm. **h** Distribution of *Tox* (C2), *Gabrg1*(C3), and *C1ql2* (C4) in tdTomato⁺ neurons from thoracic and lumbar DRG of p1 *Pv^Cre*; *Ai14* mice (mean ± SEM, n ≥ 2 animals). Source data are provided as a Source Data file.

back (erector spinae, ES) and hindlimb (gastrocnemius, GS; tibialis anterior, TA) muscles. We found that the majority of CTB⁺; *Pv*⁺ neurons connected to ES expressed *Tox*, but neither *Gabrg1* nor *C1ql2* (Fig. 4a and Supplementary Fig. 4a). Conversely, proprioceptors labeled after CTB injections in hindlimb muscles expressed *Gabrg1*, but neither *Tox* nor *C1ql2* (Fig. 4b and Supplementary Fig. 4a). Thus, retrograde labeling experiments anatomically validated the findings of transcriptome analysis.

Next, we asked whether gene expression profiles characterizing proprioceptor muscle identity at p1 were already present at earlier developmental stages. We analyzed correlation in expression of transcripts defining Ba-, Ab-, and Li-pSN identities at p1 and e15.5. As expected, we found high correlation at p1 (Fig. 4c). In addition, strong co-expression patterns of the same signature genes were also observed at e15.5 (Fig. 4d). These data indicates that molecular

features defining proprioceptor muscle identities are already present during embryonic development. To confirm this finding, we examined expression of *Tox* and *Gabrg1*, in e15.5 DRG neurons retrogradely labeled after rhodamine-dextran (RhD) injection either in back or hindlimb muscles. As previously observed for postnatal stages, we found that expression of *Tox* and *Gabrg1* in embryonic proprioceptors is predictive of their specific peripheral connectivity patterns, with *Tox* labeling RhD⁺; *Pv*⁺ back-innervating neurons and *Gabrg1* hindlimb-innervating ones (Fig. 4e, f and Supplementary Fig. 4b). Finally, we examined expression of p1 muscle-type markers (*Tox*, *Gabrg*, and *C1ql2*) in proprioceptor clusters identified at e15.5. We found that *Tox* expression characterizes three clusters (pS3, pS5 and pS6) two of which have predominant thoracic component, including the *Trpv1*⁺ neurons in pS6, that represent Ba-pSN (Fig. 1d–f, Supplementary Fig. 1k and Supplementary Fig. 7). Consistent with Li-pSN, *Gabrg1* was found

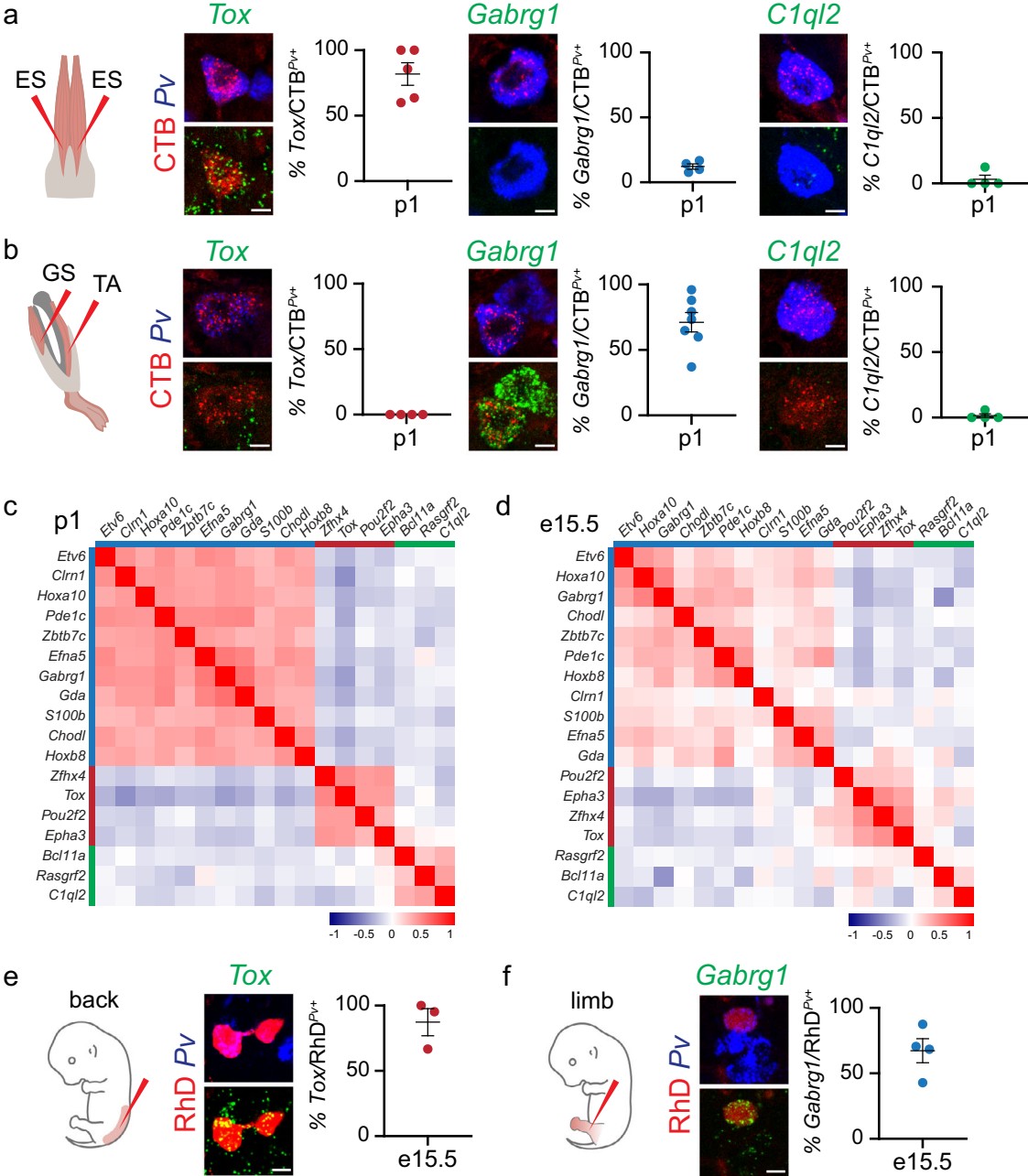

**Fig. 4 | Proprioceptors muscle-type identity emerges at early developmental stages. a, b** Representative smFISH images and quantification of *Tox* (C2), *Gabrg1* (C3), and *C1ql2* (C4) expression in *Pv*+ sensory neurons retrogradely labeled after cholera-toxin B (CTB) injection in back (**a** erector spinae, ES) and hindlimb (**b** gastrocnemius, GS and tibialis anterior, TA) of p1 wild-type mice (mean ± SEM, *n* ≥ 4 animals). Scale bar: 10 μm. **c, d** Heatmaps representing pairwise gene expression correlation values for Ba-pSN (red), Ab-pSN (green), and Li-pSN (blue) molecular signatures at p1 (top) and e15.5 (bottom; Pearson's r using logcounts). **e, f** Representative smFISH images and quantification of *Tox* (C2), and *Gabrg1* (C3), expression in *Pv*+ sensory neurons retrogradely labeled after rhodamine-dextran (RhD) injection in e15.5 back (**e**) and hindlimb (**f**) muscles of wild-type mice (mean ± SEM, *n* ≥ 3 animals). Scale bar: 10 μm. Source data are provided as a Source Data file.

in two clusters (pS2 and pS7) whose neurons originate mainly from lumbar DRG, while *C1ql2* expression characterizes pS1, the only cluster formed by a majority of thoracic neurons, thus supporting Ab-pSN identity (Fig. 1d–f, Supplementary Fig. 1k and Supplementary Fig. 7).

Altogether these data indicate that molecular profiles of proprioceptor muscle subtypes identified at p1 are already present at e15.5 and part of developmental programs arising at embryonic stages before end-organ receptor identity consolidates[14,15]. Indeed, expression of molecular signatures recently identified for Ia, Ib and II receptor subtypes do not start being correlated in our datasets until p1 (Supplementary Fig. 5).

## Ephrin-A/EphA signaling controls proprioceptor muscle targeting

The presence of molecular correlates of proprioceptor muscle character at early developmental stages suggests that signature genes defining different subtypes may be involved in the acquisition of their identities. Strikingly, the expression of *Efna5* and *Epha3* - members of the ephrin-A and EphA family of axon guidance ligands and receptors - distinguishes Ba- and Li-pSN (Figs. 3e, 4c, d, Supplementary Fig. 6a and Supplementary Fig. 7). Moreover, we found that other members of the EphA receptor family (*Epha4*, *Epha5*, and *Epha7*) are also differentially expressed in proprioceptor clusters, both at e15.5 and p1 (Fig. 5a). We

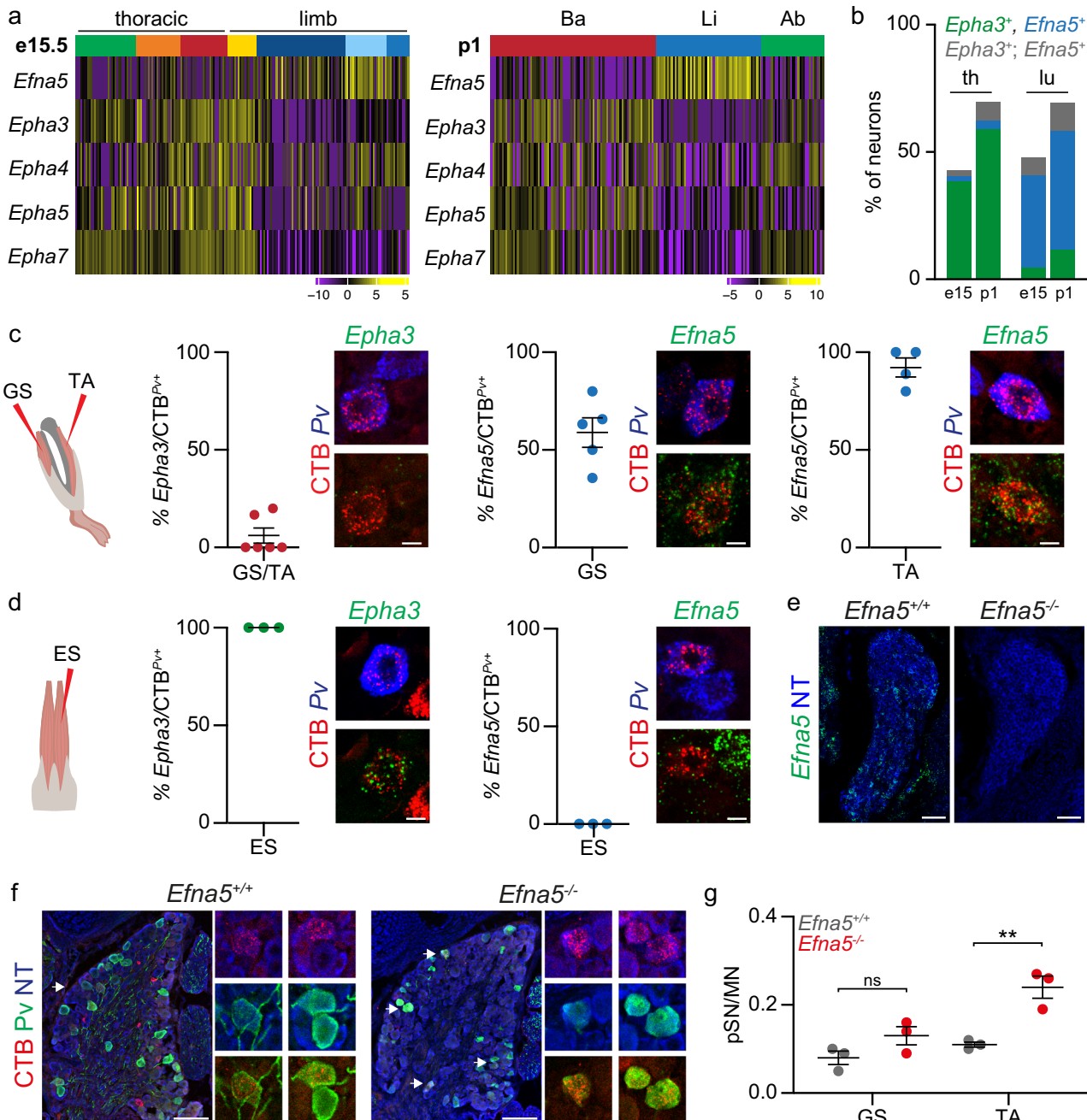

**Fig. 5 | Elimination of ephrin-A5 function erodes the specificity of muscle connectivity. a** Gene expression analysis (logcounts) of ephrin-A/EphA family members differentially expressed in proprioceptor clusters identified at e15.5 (thoracic origin: green, orange, red; lumbar origin: yellow, dark blue, light blue, and blue) and p1 (back: red; hindlimb: blue; abdominal: green). **b** Percentage of $Pv^+$ or tdTomato⁺ sensory neurons expressing *Epha3* (green), *Efna5* (blue) or *Epha3; Efna5* (gray) at thoracic and lumbar levels in either e15.5 or p1 $Pv^{Cre}$; *Ai14* mice (mean ± SEM, $n \geq 2$ animals). **c** Representative smFISH images and quantification of *Epha3* (left) and *Efna5* (center/right) expression in $Pv^+$sensory neurons retrogradely labeled after CTB injection in gastrocnemius (GS), and tibialis anterior (TA) muscles of p1 wild-type mice (mean ± SEM, $n \geq 4$ animals). Scale bar: 10 μm. **d** Representative

smFISH images and quantification of *Epha3* (left) and *Efna5* (right) expression in $Pv^+$ sensory neurons retrogradely labeled after CTB injection in the erector spinae (ES) muscle of p1 wild-type mice (mean ± SEM, $n = 3$ animals). Scale bar: 10 μm. **e** Representative smFISH images of *Efna5* expression in lumbar DRG of p1 *Efna5*^{+/+} and *Efna5*^{-/-} mice. Scale bar: 25 μm. **f** Representative images of Pv⁺; CTB⁺ sensory neurons retrogradely labeled after CTB injection in the tibialis anterior (TA) muscle of p1 *Efna5*^{+/+} (left) and *Efna5*^{-/-} (right) mice. Scale bar: 100 μm. **g** Ratio of proprioceptor (Pv⁺) per motor neuron labeled after CTB injection in the gastrocnemius (GS) and tibialis anterior (TA) muscles of p1 *Efna5*^{+/+} (gray) and *Efna5*^{-/-} (red) mice (mean ± SEM, $n = 3$ animals, two-tailed *t*-test, ns $p > 0.05$, **$p < 0.01$, p-values 0.12 GS and 0.007308 TA). Source data are provided as a Source Data file.

validated these findings in vivo by characterizing expression of *Efna5* and *Epha3* in proprioceptors labeled in *Pv; tdT* and *Trpv1; Pv; tdT* mice at e15.5 and p1 (Fig. 5b and Supplementary Fig. 6b–d). In addition, we further confirmed these data by analyzing *Efna5* and *Epha3* expression in $Pv^+$ sensory neurons retrogradely labeled after CTB injection in back and hindlimb muscles. We found that the majority of CTB⁺; $Pv^+$ neurons

connected to hindlimb muscle expressed *Efna5* but not *EphA3* (Fig. 5c). Conversely, all the proprioceptors labeled after CTB injections in ES muscle expressed *Epha3*, but not *Efna5* (Fig. 5d). These data show that *Efna5* and *Epha3* are differentially expressed in Ba- and Li-pSN neurons, suggesting a function in controlling target specificity, an intriguing possibility considering the prominent role of ephrins and their

receptors in axon guidance during development of the nervous system[28]. In order to test whether ephrin-A5 controls proprioceptor peripheral connectivity, we injected CTB in hindlimb muscles of mice lacking ephrin-A5 function (*Efna5*$^{-/-}$; Fig. 5e)[29]. First, to assess labeling specificity and whether elimination of ephrin-A5 was affecting motor neuron connectivity we examined the position and number of retrogradely labeled motor neurons. As previously reported, we did not find any significant difference in motor neuron muscle connectivity in *Efna5*$^{-/-}$ mice (Supplementary Fig. 6e, f)[30]. Next, we examined the number of retrogradely labeled proprioceptors, as well as the ratio of proprioceptor to motor neuron labeling, and found a significant increase in the number of neurons retrogradely labeled from the TA muscle, and a similar trend, although not significant, for the GS muscle, whose proprioceptors are only partially defined by *Efna5* expression (Fig. 5c, f, g and Supplementary Fig. 6g).

Thus, these data show that elimination of ephrin-A5 function erodes the specificity of hindlimb muscle connectivity and indicate that the molecular signatures of muscle subtypes comprise programs controlling defining features of proprioceptor muscle-type identity.

## Discussion

This work defines the molecular signatures underlying proprioceptor subtypes defined by their muscle connectivity. We found that molecular distinctions emerge during embryonic development before the onset and consolidation of receptor character and comprise programs that control the specificity of muscle connectivity. These findings set the stage for defining the mechanisms controlling the acquisition of proprioceptor identity at a single muscle level and the generation of a toolbox for analyzing the physiological roles of proprioceptor subtypes and define the contribution of sensory feedback from different muscle groups in the control of movement and the generation of the sense of body position in space.

We identified and validated molecular signatures for proprioceptor innervating cardinal muscle groups: back (*Tox*, *Epha3*), abdominal (*C1ql2*), and hindlimb (*Gabrg1*, *Efna5*). Markers for back and abdominal subtypes (*Tox* and *C1ql2*) account for almost the entire proprioceptor population in thoracic DRG (~ 88%; Fig. 3g), thus indicating that our approach comprehensively captured most of the neurons innervating muscles at trunk level. In contrast, both *Gabrg1* (~46%) and *Efna5* (~66%) alone only capture about half of limb-innervating proprioceptors each, but together they account for about 75% of Pv$^+$ sensory neurons in lumbar DRG (Figs. 3g and 5b and Supplementary Fig. 3j). The anatomical complexity of limbs, comprising 39 different muscles in the mouse hindlimb[31], is consistent with a model requiring a combination of multiple molecules in order to represent the whole compartment. The presence of multiple clusters associated with general back and hindlimb identities at e15.5 indicate that their molecular makeup may already capture features of more fine-grained identities defined according to specific anatomical (i.e.: rostral vs. caudal back; dorsal vs. ventral limb) or functional (i.e.: synergist vs. antagonist) characteristics. However, our relatively small dataset makes the interpretation of these clusters difficult and increasing the number of cells analyzed will be necessary in future studies to shed light on the nature of these neurons.

Upon acquisition of a generic proprioceptor fate[32], sensory neurons mature to develop functional features defined by their muscle and end-organ receptor identities. First, sensory axons navigate peripheral targets and innervate mechanoreceptive end-organs with precise ratios and distributions according to the biomechanical requirements of the innervated muscle[33]. In addition, each proprioceptor subtype needs to establish specific sets of connections with multiple neural targets in the central nervous system in order to relay sensory feedback to motor circuits controlling the activity of relevant muscles[6,34]. Our data support a model where proprioceptor muscle identity emerges as part of an embryonic genetic program controlling

connectivity to its central and peripheral targets that is refined at later stages to include aspects of receptor-type character (MS and GTO), such as distinct physiological properties, whose diversification is influenced by neuronal activity[15]. In support of this view, signatures of proprioceptor muscle-type identities are clearly evident from e15.5, while group Ia, Ib, and II molecular profiles have been shown to emerge later and consolidate during postnatal development[14,15,35]. Accordingly, molecular correlates defining receptor identity are not immediately evident in the muscle-type profiles we identified at e15.5, but start emerging at p1. A notable exception is represented by *Tox* and *Chodl*, which have been previously proposed to represent markers of two groups of type II afferents (II$_2$ and II$_4$) at early postnatal stages[15]. These molecules define back (*Tox*) and hindlimb (*Chodl*) muscle subtypes in our analysis. Interestingly, groups II$_2$ and II$_4$ were found to be enriched in DRG at thoracic and lumbar levels respectively, thus confirming our results and indicating that the diversity observed in type II proprioceptors may already include signatures of muscle-type identity[15]. Altogether, these observations suggest that "receptor" features become superimposed to "muscle" character already present since early development in order to generate the final functional subtype identity. Future studies building on these findings bear the promise to define the developmental processes controlling proprioceptor specification from general proprioceptive fate determination to the acquisition of muscle-type identity and maturation of physiological characteristics at receptor level.

The specificity with which proprioceptors innervate respective muscle targets in the periphery and synaptic partners in the central nervous system provides the circuit basis for the function of spinal sensorimotor circuits[36]. Our data shows that the ephrin-A/EphA family of axon guidance molecules is an important regulator of proprioceptor peripheral connectivity. We found that differential expression of *ephrin-A5* and four EphA receptors (*EphA3*, *EphA4*, *EphA5*, and *EphA7*) delineate a distinction between hindlimb- and abdominal/back-projecting proprioceptors, and perturbation of ephrin-A5 function leads to an erosion in the specificity of muscle connectivity. The phenotype indicates that Efna5 may be part of a developmental program controlling the precision of muscle innervation. Ephrin signaling is known to have important roles in the guidance of somatosensory and motor axons to their peripheral targets[28]. It has been shown that at early embryonic stages nascent sensory axons track along motor axons en route to their peripheral targets and trans-axonal interactions control navigation of sensory neurons to axial targets[37]. In particular, interactions between EphA3/4 in motor axons and ephrin-A2/A5 in somatosensory axons have been shown to control innervation of the epaxial compartment by sensory neurons. In their absence, epaxial sensory nerves are re-routed to hypaxial targets[38]. We observed a significant increase in the number of proprioceptors innervating the tibialis anterior muscle in mice lacking ephrin-A5, indicating that excessive limb muscle innervation might result from mistargeting of axons originally directed to another muscle compartment whose identity remain elusive. However, we did not observe any difference in the connectivity to a representative back muscle, as the number of pSN retrograde labeled after CTB injection in the erector spinae muscle of Efna5$^{-/-}$ and control mice was unaffected (Supplementary Fig. 6h-k). Ephrin-A/EphA signaling also controls the choice of limb-innervating motor neurons to invade either the dorsal or ventral half of the limb mesenchyme and could influence muscle by muscle dependence of proprioceptive axon innervation specificity[30,39,40]. Because of the intricacy of ephrin-Eph signaling[28], it will be necessary to carefully analyze the expression pattern and function of different ligands, receptors, and coreceptors in order to define the molecular logic governing guidance of proprioceptors to their specific peripheral targets.

The importance of proprioceptive sensory feedback in motor control is clearly evident in mouse models where proprioceptor

development or function is perturbed. Degeneration of muscle spindles in Egr3 mutant mice result in ataxia and, similarly, loss of most proprioceptors in absence of Runx3 function results in severe coordination phenotypes[41–43]. Moreover, elimination of the mechanosensory transduction channel Piezo2 in proprioceptors leads to severely uncoordinated body movements and limb positions[44]. Despite the critical role of proprioception for the generation of coordinated movement, it is still not understood how proprioceptive feedback from different muscles and receptor subtypes integrates with motor commands and other sources of sensory input to adjust motor output and generate the sense of body position in space[45,46]. This is mainly due to the fact that behavioral studies have been hampered by the lack of tools allowing precise access to different functional subtypes of proprioceptors. The identification of molecular signatures for proprioceptor muscle subtypes opens the way for the generation of genetic and viral tools to selectively access distinct channels of proprioceptive information and bears the promise to determine their roles in motor control.

## Methods

### Animal experiments
All animal experiments were performed in compliance with the German Animal Welfare Act and approved by the Regional Office for Health and Social Affairs Berlin (LAGeSo) under license numbers G0148/17 and G0191/18.

### Animal models
Mice were housed in standardized cages under 12 h light-dark cycle with food and water *ad libitum*. For this study the following mouse lines were used $Pv^{Cre\ 27}$, $Pv^{Flp\ 25}$, $Pv^{tdTom\ 17}$, $Trpv1^{Cre\text{-}Basbaum\ 23}$, $Trpv1^{Cre\text{-}Hoon\ 22}$, $Ai14^{24}$, $Ai65^{25}$, and $Efna5^{-/-\,29}$.

### Single-cell isolation
Dorsal root ganglia were dissected separately from thoracic (T1-T12) and lumbar (L1-l5) segments and collected in F12 medium with 10% FHS (Fetal horse serum) on ice. Next, DRG were incubated in F12/FHS with 0,125% collagenase (Sigma C0130) for 1 h (p1) or 30 min (e15.5) at 37 °C. After 3 washes with PBS DRG were transferred to 0,25% trypsin solution (Gibco 15050-065) and incubated for 15 min at 37 °C. Afterwards, DRG were mechanically triturated using a fire polished Pasteur pipette until a homogenized solution was visible followed by a centrifugation step at 200 × *g* for 10 min. The final cell pellet was resuspended in HBSS (10 mM HEPES, 10 mM Glucose) and the resulting cell suspension either applied to fluorescence-activated cell sorting (FACS) (e15.5) or manual cell picking under an inverted fluorescent microscope (p1). Single tdTomato⁺ cells were sorted into individual wells containing lysis buffer and stored at −80 °C until further processing.

### Single-cell RNA sequencing
For cDNA library preparation the CEL-Seq2 protocol was used as previously described[18]. We sequenced 960 cells (480 from T1-T12 and 480 from L1-L5) at e15.5 and 576 (96 thoracic and 96 lumbar from $Pv^{Cre}$; $Ai14$; 96 thoracic and 96 lumbar from $Trpv1^{Cre\text{-}Basbaum}$; $Pv^{Flp}$; $Ai65$; 96 thoracic and 96 lumbar from $Trpv1^{Cre\text{-}Hoon}$; $Pv^{Flp}$; $Ai65$) at p1. The libraries were sequenced on an Illumina NextSeq500 platform with high-output flow cells by the Next Generation Sequencing Core Facility of the Max-Delbrück Center for Molecular Medicine.

### Single-cell analysis
For both datasets (e15.5 and p1) we used the scruff v1.4.0 package (R package version 1.12.0) to demultiplex, map, and generate count matrices. Then, we evaluated each dataset statistics using Scater v1.14.6R package. To increase the quality of the experiments, we individually removed low-quality cells based on low total gene counts (> quantile 0.3), low gene abundance (> quantile 0.3), and high mitochondrial gene values cells (<quantile 0.75). 519 out of 960 e15.5 cells and 244 out of 576 p1 cells passed quality control criteria. After log-normalization, we used the scran v1.14.1 "buildKNNGraph" and "cluster_walktrap" functions with default parameters to define each dataset cell populations and subclusters. Finally, we assigned gene markers to each population using "findMarkers" function from the scran with default parameters. For single-cell analysis R v3.6.2 environment was used to generate the results, statistical analysis and graphical evaluation of the datasets.

### Dissection and tissue processing
Postnatal mice were anesthetized by intraperitoneal injection of 120 mg/kg ketamine and 10 mg/kg xylazine and transcardially perfused with PBS and 4% PFA in 0.1 M phosphate buffer. To expose the spinal cord a ventral laminectomy was performed and the tissue post-fixed O/N in 4% PFA at 4 °C. The next day tissue was washed three times with ice-cold PBS and transferred to 30% sucrose in PBS for cryoprotection at 4 °C O/N. Tissue was embedded in Tissue-Tek OCT embedding compound and stored at −80 °C. 16 µm tissue sections for immunohistochemistry were acquired at a cryostat, dried for 1 h and either directly used or frozen at −80 °C.

### Immunohistochemistry and fluorescent in situ hybridization
For immunohistochemistry dry tissue sections were washed for 10 min with PBS followed by another 10 min incubation of 0.1% Triton-X-100 in PBS (0.1% PBX) for permeabilization. The following primary antibodies were diluted in 0.1% PBX and incubated O/N at 4 °C: Ch-anti-Pv (1:5000, generous gift from Susan Brenner-Morton), Goat-anti-ChAT (1:200, Millipore #AB144P), GP-anti-vGluT1 (1:5000, generous gift from Susan Brenner-Morton), Rb-anti-dsRed (1:1000, Takara #632496) and Rb-anti-RFP (1:500, Rockland #600-401-379). Next, slides were washed three times for 5 min with 0.1% PBX followed by secondary antibody/NeuroTrace incubation for 1 h at room temperature (RT). Secondary antibodies (Jackson Immuno Research Laboratories) and NeuroTrace (Life Technologies) were diluted in 0.1% PBX as following: Cy3, Alexa488 (1:1000), Cy5 (1:250), and NeuroTrace (1:250). After staining with secondary antibodies slides were washed three times with 0.1% PBX and subsequently mounted with Vectashield antifade mounting medium. For fluorescent in situ hybridization the RNAscope Multiplex Fluorescent Kit v2 (ACDBio) with a modified manufactures protocol was used. Tissue sections were acquired as described above. Sections were dried, fixed with ice-cold 4% PFA in PBS for 15 min and dehydrated in a series of 50%, 70% and 100% ethanol for 5 min each. Afterwards, sections were treated with hydrogen peroxide solution for 15 min at RT to block endogenous peroxidase activity followed by another wash with 100% ethanol for 5 min. Next either Protease IV (postnatal tissue) or Protease III (embryonic tissue and sections from CTB tracing experiments) was applied for 30 min at RT. After three washes with PBS, probes were applied and hybridization performed in a humified oven at 40 °C for 2 h. The following probes were used in this study: Mm-Epha3-C1, Mm-Tox-C1, Mm-C1ql2-C1, Mm-Efna5-C2, Mm-Trpv1-C2, Mm-Pvalb-C2, Mm-Pvalb-C3, Mm-Gabrg1-C3, and Mm-Runx3-C3. Following hybridization, amplification was performed using Amp1, Amp2, and Amp3 each for 30 min at 40 °C. For detection each section was treated sequentially with channel specific HRP (HRP-C1, HRP-C1, HRP-C3) for 15 min, followed by TSA mediated fluorophore (Akoya Bioscience, Opal 520, Opal 570, and Opal 690) binding for 30 min and final HRP blocking for 15 min (all steps at 40 °C). When necessary additional immunostaining was performed as described above. Images were acquired with a Zeiss LSM800 (ZENblue v 3.4.8) confocal microscope. Cell bodies (evaluated by Nissl staining) colocalizing with ≥5 puncta were counted positive. For quantification ImageJ2 v 2.3.0/153f and Adobe Photoshop v 25.4.1 (count tool) were used.

## Tissue clearing and light-sheet microscopy

Mice were anesthetized and transcardially perfused as described above. Afterwards, spinal cord and/or DRG were extracted after ventral laminectomy and postfixed in 4% PFA for 2 days at 4 °C. DRG were kept separately and embedded into 1% low melt agarose in OptiPrep (Sigma) after post fixation. Tissue clearing was performed as previously described with modifications[47]. In short, tissue was transferred to CUBIC1 (25 wt% Urea, 25 wt% N,N,N′,N′-tetrakis(2-hydroxypropyl) ethylenediamine, 15 wt% Triton X-100) and incubated at 37 °C shaking. Every other day CUBIC1 solution was exchanged until tissue appeared transparent (spinal cord ~4 days, DRG ~1–2 days). Afterwards, samples were washed for 1 day with PBS at RT, refractive index matched with EasyIndex (LifeCanvas Technologies) at 37 °C and imaged with the ZEISS Light-sheet Z.1 (ZENblack v 3.1). For image analysis and video rendering Arivis Vision4D v3.5.1 (Arivis AG) and Imaris v9.8.0 (Oxford Instruments) was used.

## Retrograde labeling of proprioceptors and motor neurons

For retrograde labeling of p1 proprioceptors, mice were anesthetized with isoflurane and a small incision on the skin was made to expose the muscle of interest. 50 nl of a 1% solution of Alexa555-conjugated CTB (Life Technologies) was injected with a glass capillary into the desired muscles. Animals were sacrificed and perfused after 3 days. For retrograde labeling of e15.5 proprioceptors, embryos were dissected in ice-cold artificial cerebrospinal fluid and pinned down. Next, skin from limb or back muscles was removed and 20% rhodamine dextran (Life Technologies) injected into the desired muscle using a pulled glass capillary. Afterwards, embryos were incubated in circulating oxygenated artificial cerebrospinal fluid (5% $CO_2$, 95% $O_2$) for 6 h at 27 °C and fixed with 4% PFA.

## Statistics and reproducibility

Details for statistical analysis and number of samples are indicated in figure legends. Significance for $t$-tests was defined as $*p < 0.05$; $**p < 0.01$; $***p < 0.001$.

For immunostaining, smFISH and tracing experiments, replicates were performed in $n \geq 3$ animals even in cases where only representative images were presented in the figure.

## Reporting summary

Further information on research design is available in the Nature Portfolio Reporting Summary linked to this article.

# Data availability

Single-cell-transcriptome data is accessible at the NCBI GEO repository, accession code: GSE190605. Source data are provided with this paper.

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

## Acknowledgements
We thank Liana Kosizki for technical support, Mathias Richter and the MDC Advanced Light Microscope facility for assistance with light-sheet microscopy. We are grateful to Martyn Goulding and Mark Hoon for sharing mouse lines. We thank Susan Brenner-Morton for sharing antibodies. We thank Robert Manteufel, Ilka Duckert, and Florian Keim for animal care. We are grateful to Dario Bonanomi for helpful discussions; Nikos Balaskas, Joriene de Nooij, and members of the Zampieri laboratory for insightful comments on the manuscript. N.Z. was supported by DFG grant ZA 885/1-2; G.G. by Helmholtz (VH-NG-1153), KWF (NKI-2014-7208), and ERC (714922).

## Author contributions
Conceptualization, S.D. and N.Z.; Investigation, S.D., C.C., K.S., L.R., and E.D.L.; Formal analysis, S.D., C.C., L.R. and G.G.; Writing – Original Draft, S.D. and N.Z.; Writing – Review and Editing, S.D., C.C., K.S., L.R., E.D.L., C.B., G.G., and N.Z.; Supervision, C.B., G.G., and N.Z.

## Funding

## Competing interests
The authors declare no competing interests.
