## [Peer Review File · Nature Communications]

Molecular identity of proprioceptor subtypes innervating different muscle groups in miceReviewers' comments:

Reviewer #1 (Remarks to the Author):

This manuscript used single cell transcriptomics to investigate the molecular identity of proprioceptors innervating different muscle groups in the body: axial muscles of the back of the trunk (epaxial), abdominal muscles (hypaxial) and limb muscles. This information is important to understand signaling involved in developing their specific connections in the periphery, as well as their central connections, as well as genes that direct specification and differentiation of proprioceptors with specific muscle identities. The distribution of proprioceptors to specific muscles and central pathways is critical for the system to work. The information described lead to the discovery of specific genes from proprioceptors targeting each of these muscle groups, but hypothesis testing of possible functions for some of these genes was more confusing. A second advance in this paper is that through detailed comparison of transcriptomic data at E15.5 and P1, that is before and after proprioceptors connect to peripheral and central targets, they conclude that molecular identity associated to muscle projection is established early in development. This is not too surprising since it is not dissimilar to motoneurons and it makes sense that guidance systems are in place to direct axons to the correct body locations. Recent studies also suggest that functional proprioceptive identity into the three major classes of proprioceptors occurs later in development (Ia, Ib and II). Here there is also a parallel with motoneurons in which functional maturation into different motor units is a relative developmental action. Therefore, it seems that similar developmental sequences as found in other parts of the motor systems occurs during proprioceptors development: first specification of proprioceptive identity, second muscle identity, third functional differentiation. The manuscript fills an important gap in the second phase that was previously unknown. Most importantly the study generates key data sets for future genetic manipulations to probe the system. Therefore, although this study is currently mostly at the discovery/descriptive phase, I believe it is still of high significance.

The sets of genes differentially expressed in proprioceptor projecting to each of three classes of muscles includes some interesting surprises. For example, Trpv1 is present in some epaxial proprioceptors. However, the most confusing part of this manuscript is their only attempt to ascribe a function to one differentially expressed gene. Efna5 is not present in epaxial proprioceptors but it is in >90% of proprioceptors project to the limb tibialis anterior (TA) muscle. Genetic deletion of Efna5 results in increased numbers of TA projecting proprioceptors and no change in gastrocnemius (GS) proprioceptors. The results are interpreted as a repulsive action of Efna5 on epaxial proprioceptors that are now projecting specifically to the TA avoiding the GS, but if this is the case why is it in almost all TA axons and why is not repulsive to them? Moreover, can the authors strengthen this interpretation by confirming lack of muscle spindles in lumbar epaxial muscles in Efna5 knockouts? The results also open a number of more punctual questions that should be clarified such that interpretations become clearer.

1. Figure 1 transcriptomic data show 7 groups of proprioceptors, 4 of which were associated with lumbar (Hoxc10+) regions. What is the significance of these 4 groups? What are their differential genes?
2. Similarly, there are three groups that are enriched at thoracic levels, but some within these groups are lumbar (Hoxc10+) cells. What is the significance of having 3 different groups of mixed thoracic and lumbar proprioceptors with higher internal similarity (independent of origin) than among the groups?
3. If Trpv1 proprioceptors are most significant for epaxial musculature, why is it that extended data figure 2 presents only negative data in abdominal hypaxial musculature? What is the percentage of Trpv1 PV tdT spindles to all spindles in epaxial musculature? The present graph in e with just zeros is not informative.
4. In figures 3 and beyond please make sure is clear when the marker labeling is coming from ICC or RNAscope. It is at present not clear. In materials and methods please make sure all reagent used are adequately described.
5. Ephrin data. Was Efna5 tested in retrogradely labeled ES proprioceptors and EphA3 is GS and TA proprioceptors? That will be a nice confirmation of the transcriptomic data on Ephrins and their receptors. At present it seems these markers were tested only in the expected proprioceptors.
6. Why would a marker of type II afferents (tox) be present in 100% of epaxial proprioceptors many of

which have characteristic central projections of Ia afferents, not type IIs?

7. The data shows identity demonstrated by unique genes for epaxial and hypaxial musculature, but not such unique genes exist for hindlimb, does that mean the limb proprioceptors are best defined by combinatorial criteria? Or just by exclusion of epaxial and hypaxial genes? Can this information be extracted from the data?

Reviewer #2 (Remarks to the Author):

In this paper the authors have identified muscle target specific markers (epaxial, limb, and hypaxial) in proprioceptors that are present before proprioceptor subtype (Ia/II/Ib) development. The authors also identified the axon guidance molecules from the ephrin-A/EphA family as important regulators of proprioceptor guidance during development. In ephrin-A5 knockout mice, increased proprioceptor innervation was found in TA but not GS muscles. The identified epaxial, limb, and hypaxial specific markers should allow for future studies dissecting the role of proprioceptors innervating different muscle types. While more specific markers to different muscle groups will be helpful in those future studies, these identified markers are an excellent start.

Interestingly the authors also identified TrpV1 as a label for a subset of epaxial innervating proprioceptors. To my knowledge this is the first identification of TrpV1 in proprioceptors. In the discussion it would be helpful to discuss the potential significance of the presence of TrpV1 and whether the authors have any ideas about what role it might be playing.

Overall, this was a well done paper that identifies genetic markers that will be of use to other researchers interested in dissecting the role of proprioceptors from different muscle groups. The authors have used reasonable methods to confirm their putative markers. I can find no major flaws in the study and have only minor questions/suggestions.

Minor Comments:

On Fig 1 b & d it would be helpful to describe or show in the figure legend what, if anything, the different colors mean.

Did you look at whether there were changes in epaxial innervation in the ephrin-A5 KO mice? In the discussion, you speculate that the mistargeted proprioceptors in the TA muscle may have been intended for epaxial muscles. If so, wouldn't you expect decreased innervation in epaxial muscles as well?

Reviewer #3 (Remarks to the Author):

Comments on Dietrich et al.

This manuscript by Dietrich et al. described molecular heterogeneity of pSNs at two developmental stages corresponding to E15.5 and P1, from thoracic and lumbar levels. For this, the authors are using scRNAseq with Cell-seq2 method and show that DRG cells from the above regions can be clustered into known cell types and then subclustered into proprioceptive neurons clusters, which they suggest might represent muscle-group specificity. They further attempt to correlate those subgroups of pSNs to axon guidance molecules, and tested Ephrin A5 (EfnA5) expression as a requirement for pSN subgroups axon targeting. Beyond methodological considerations (see below), a significant problem with this study is that it does not present a demonstration of any pSN subtype identities.

Major concerns:

Omic:

While the study is presented as based on scRNAseq data, no scRNAseq data analysis have been performed but one in fig3 e and is very limited (as it is, most of the genes cited throughout the manuscript seem to be selected by the authors but do not arise from RNAseq analysis per se).

- data on gene expression per se are missing throughout the manuscript, including those related to the efficiency (gene coverage) of their method and to the number of cells included in the clustering (see below). Also, the average of gene expression per cell and per cluster is missing.

- the manuscript contains only few heatmaps showing selected known markers and one heatmap of top expressed genes in fig3e. The expression unit is not specified.

- The actual number of pSNs clustered and used for further analysis, for E15.5 and for P1, is unknown. These numbers seem very low, considering the panel 1d for instance. Importantly, clustering in Fig.1d seems arbitrary; no analysis is performed, no molecular markers (or top expressed genes or clusters 'exclusive genes expression) are shown, and no confirmation is provided in situ, yet the authors claim those are related to muscle specific groups and name them (i.e. fig3 and l181-187, the cluster assignment is arbitrary).

- Why the authors change from E15.5 to p1 scRNAseq is unclear and p7 for anatomical analysis of the spinal cord and muscles.

Inconsistencies:

1) it is unclear how the authors obtained so many dividing cells (obviously not neuronal) of unknown identity in their cell sorting-seq method in fig.1, if using PVcre;tomato to trace pSNs (PV is not expressed in progenitors).

2) It is not described how the authors came to focus on Trpv1, which, in contrast to what the authors write, seems in Fig. 1g to be enriched only in a part of pC6, of lumbar origin, and in a minority of pSNs in situ (Fig. 1i, only 5% at thoracic level).

3) The use of many shortcuts and erroneous statements to claim generalities and support their conclusions.

- For instance, the molecular identity of pSN subtypes has been demonstrated already, at embryonic, postnatal and adult stages (Oliver et al., 2021, Nat Commun; Wu et al., 2021, Nat Commun). Dietrich et al. however write that Ia, Ib and II subtypes are not existing prior to postnatal stages, citing Oliver et al. and Wu et al. as references. Wu et al. however clearly demonstrated Ia, Ib and II pSNs at E16.5. It is therefore unclear why Dietrich et al. claim the opposite.

- In line with the previous comment, the authors claim Efna5 is not necessary for motor neurons muscle connectivity, citing Bonanomi et al. (2012, Cell), but Bonanomi et al. actually demonstrated in this study that it is essential for proper motor neuron muscle innervation, and sensory axon outgrowth depends on correct motor axon patterning (Wang et al., 2011, Neuron; Wang et al., 2014, Development). Moreover, Wang et al., (2011) demonstrated that Efna5 is expressed in epaxial sensory axons (the opposite in Dietrich et al.) at the time of axon outgrowth and is necessary for sensory axon innervation. Dietrich et al. only analyzed expression of Efna5 at a much later stage (P1, in Trpv1 positive pSNs), which does not correspond to development of innervation (<E14).

5) How the clusters found in Dietrich et al. relate to Ia, Ib and II pSNs is not investigated. Indeed, the authors extracted and sequenced all DRGs neurons expressing PV which for the proprioceptors includes all major pSN subtypes. In fact, the datasets of Oliver et al. and of Wu et al. (including at E16.5) are available online and can be used to understanding further specificity beyond Ia, Ib and II pSNs during early development.

6) Generalized markers to subtypes while their expression only covers part of cluster and groups of neurons; for instance, Trpv1 is used for epaxial-pSNs while Trpv1 is expressed only in 5% of thoracic pSNs. Also, the term "subtypes" cannot be used as there is no data indicating that the clusters they found represent functionally different subtypes.

7) To analyze the molecular regulation of muscle-by-muscle or muscle groups innervation by pSNs, scRNAseq should be at least performed during the period of innervation. The phenotype they found in the Efna5 null mice could be due to defect in motor axons, or earlier defects.

8) The mutant data points distribution for the number of MN labelled by injection of CTB in 2 different muscles GS and TA seem completely identical.

9) CTB tracing present multiple advantages, one of those is that the positively traced cells are always very bright and easy to identify; in the panel fig5e, the fluorescence observed is uncommon with a lot of background (see lower panel), which is very similar to the fainted labelling in pointed cells.

Other concerns:

- novelty wise, molecular markers such as *Tox*, *Chodl*, *EfnA5* and *Gabrg1* were already known markers for pSN rostro-caudal distribution (Wu et al. 2021 Fig3).
- Moreover, a previous study in 2016 by Poliak et al. identified a series of muscle-type specific pSN markers using Affymetrix screening. It is surprising that the authors, here, did not attempt to compare these reference data.
- the limited number of muscles studied might prevent generalized statement outside the discussion paragraph.

Re: Point-by-point response to Reviewers NCOMMS-22-11523A-Z

Reviewer #1:

We thank Reviewer #1 for highlighting the significance of the work.

*This manuscript used single cell transcriptomics to investigate the molecular identity of proprioceptors innervating different muscle groups in the body: axial muscles of the back of the trunk (epaxial), abdominal muscles (hypaxial) and limb muscles. This information is important to understand signaling involved in developing their specific connections in the periphery, as well as their central connections, as well as genes that direct specification and differentiation of proprioceptors with specific muscle identities. The distribution of proprioceptors to specific muscles and central pathways is critical for the system to work. **The information described lead to the discovery of specific genes from proprioceptors targeting each of these muscle groups**, but hypothesis testing of possible functions for some of these genes was more confusing. **A second advance in this paper is that through detailed comparison of transcriptomic data at E15.5 and P1, that is before and after proprioceptors connect to peripheral and central targets, they conclude that molecular identity associated to muscle projection is established early in development.** This is not too surprising since it is not dissimilar to motoneurons and it makes sense that guidance systems are in place to direct axons to the correct body locations. Recent studies also suggest that functional proprioceptive identity into the three major classes of proprioceptors occurs later in development (Ia, Ib and II). Here there is also a parallel with motoneurons in which functional maturation into different motor units is a relative developmental action. Therefore, it seems that similar developmental sequences as found in other parts of the motor systems occurs during proprioceptors development: first specification of proprioceptive identity, second muscle identity, third functional differentiation. **The manuscript fills an important gap in the second phase that was previously unknown. Most importantly the study generates key data sets for future genetic manipulations to probe the system. Therefore, although this study is currently mostly at the discovery/descriptive phase, I believe it is still of high significance.***

*The sets of genes differentially expressed in proprioceptor projecting to each of three classes of muscles includes some interesting surprises. For example, Trpv1 is present in some epaxial proprioceptors. However, the most confusing part of this manuscript is their only attempt to ascribe a function to one differentially expressed gene. Efn5 is not present in epaxial proprioceptors but it is in >90% of proprioceptors project to the limb tibialis anterior (TA) muscle. **Genetic deletion of Efn5 results in increased numbers of TA projecting proprioceptors and no change in gastrocnemius (GS) proprioceptors. The results are interpreted as a repulsive action of Efn5 on epaxial proprioceptors that are now projecting specifically to the TA avoiding the GS, but if this is the case why is it in almost all TA axons and why is not repulsive to them?***

- It is unfortunately difficult to interpret the data on TA connectivity because of the complexity of Eph/ephrin interactions where forward and back signaling from multiple partners (proprioceptors, other somatosensory neurons, motor neurons, mesenchyme, muscles, etc.) during development has the potential to result in several attractive or repulsive outcomes that can explain the phenotype observed in the global *Efna5*^{-/-} mice. The data show that also at the level of GS there is a trend toward increased connectivity in *Efna5*^{-/-} mice that however is not significant (Fig. 5g). This could be due to the fact that for GS only a bit more than 50% of the connected pSN are *Efna5*⁺ and therefore the effect may not be as penetrant as in the case of the TA (almost all the TA connected pSN are *Efna5*⁺; Fig. 5c). Understanding the logic with which ephrin signaling controls pSN muscle targeting is an important question, but it goes beyond the scope of the current manuscript. In order to comprehensively address this question it would require an ad hoc study carefully analyzing the outcomes of multiple knock-out mice, including conditional elimination of *Efna5* from back-pSN, motor neurons and targets in the periphery, as well as the effect of potential Eph receptors.

Moreover, can the authors strengthen this interpretation by confirming lack of muscle spindles in lumbar epaxial muscles in Efna5 knockouts?

- We have explored this possibility. We checked connectivity to back muscles in *Efna5*^{-/-} mice to test whether there is a decrease that matches the increase in the innervation of limb muscles. However, retrograde labelling experiments after CTB injection in ES shows that there are no differences in the number of pSN labeled in *Efna5*^{-/-} mice. We have added these findings to the manuscript (Extended Data Fig.6h-k).

The results also open a number of more punctual questions that should be clarified such that interpretations become clearer.

1. Figure 1 transcriptomic data show 7 groups of proprioceptors, 4 of which were associated with lumbar (Hoxc10+) regions. What is the significance of these 4 groups?

- We believe that these groups may represent more fine-grained distinctions between hindlimb muscles; maybe reflecting functional (i.e.: flexor vs extensor) or anatomical (i.e.: muscles operating at different joints) differences, we have added new data to highlight these aspects. Indeed, analysis of *Gabrg1* and *Efna5* expression (markers of general hindlimb muscle identity at p1) at e15.5 seem to suggest the existence of multiple hindlimb clusters that can be already differentiated by their differential expression. *Gabrg1* is found in “lumbar” clusters pC2 and pC7 while *Efna5* predominantly in “lumbar” cluster pC4 (Extended Data Fig. 1k and Extended Data Fig. 6a).

What are their differential genes?

We have now included more information about transcriptomic analysis at e15.5 including differential gene expression analysis. Please see Fig. 1g and Extended Data Fig. 1.

2. Similarly, there are three groups that are enriched at thoracic levels, but some within these groups are lumbar (Hoxc10+) cells. What is the significance of having 3 different groups of mixed thoracic and lumbar proprioceptors with higher internal similarity (independent of origin) than among the groups?

- As in the case for the e15.5 “lumbar” clusters, we favor the hypothesis that these mixed “thoracic/lumbar” represent different subtypes of back muscles. Indeed, expression of *Tox* - marker of general back muscle identity at p1 – is found in all of these clusters (pC2, pC5 and pC6; Extended Data Fig. 1k).

3. If Trpv1 proprioceptors are most significant for epaxial musculature, why is it that extended data figure 2 presents only negative data in abdominal hypaxial musculature?

- In Fig. 2 we show that intersectional genetic labeling using *Trpv1-Cre* in combination with *Pv-Flp* can be used to specifically label epaxial proprioceptors (those connected to back muscles) and, to further support this finding and the specificity of the genetic strategy, in Extended Data Fig. 2 we showed that hypaxial proprioceptors (connected to abdominal muscles) are not labelled. We now modified Extended Data Fig. 2 to include more pictures of epaxial musculature labelled in the *Trpv1-Cre; Pv-Flp* reporter mice and relative quantifications.

What is the percentage of Trpv1 PV tdT spindles to all spindles in epaxial musculature? The present graph in e with just zeros is not informative.

- We observe labeling of nearly all MS and more than half GTO in the erector spinae muscle, a representative epaxial muscle (Fig. 2f). As suggested by the Reviewer we eliminated the graph reporting the number of MS labelled in abdominal (hypaxial) muscles, that is zero as they are not labelled.

4. In figures 3 and beyond please make sure is clear when the marker labeling is coming from ICC or RNAscope. It is at present not clear. In materials and methods please make sure all reagent used are adequately described.

- Italics font identify RNAscope while roman font ICC.

5. Ephrin data. Was *Efna5* tested in retrogradely labeled ES proprioceptors and *Epha3* is GS and TA proprioceptors? That will be a nice confirmation of the transcriptomic data on Ephrins and their receptors. At present it seems these markers were tested only in the expected proprioceptors.

- We have added these experiments for both *Tox/Epha3* and *Gabrg1/Efna5* that represent markers for back- and limb-proprioceptors respectively. As expected *Tox* and *Epha3* (Back-pSN markers) are not found expressed in *Pv⁺/CTB⁺* pSN labelled after limb muscle injection and, conversely, and *Efna5* (Limb-pSN markers) are not found expressed in *Pv⁺/CTB⁺* pSN labelled after back muscle injection (Fig. 4a, b and Fig. 5c, d). In addition, we also found that, *C1ql2* a marker for abdominal-pSN is not expressed in neither back- or limb-pSN retrogradely labelled from back (ES) or limb muscles (GS/TA), thus further confirming the specificity of the identified markers for back, abdominal and limb projecting pSN (Fig. 4a, b).

6. Why would a marker of type II afferents (*tox*) be present in 100% of epaxial proprioceptors many of which have characteristic central projections of Ia afferents, not type IIs?

- Our data support the model that *Tox* is primarily an epaxial marker. We discuss this in lines 357-364.

7. The data shows identity demonstrated by unique genes for epaxial and hypaxial musculature, but not such unique genes exist for hindlimb, does that mean the limb proprioceptors are best defined by combinatorial criteria? Or just by exclusion of epaxial and hypaxial genes? Can this information be extracted from the data?

- The presented data already hints to the fact that multiple markers are required to represent the totality of hindlimb muscles (*Efna5* and *Gabrg1* together cover the majority of lumbar proprioceptors at p1; Extended Data Fig. 3j). We discussed this important point in lines 329-339.

Reviewer #2:

We thank Reviewer #2 for appreciating the quality and importance of our data.

In this paper the authors have identified muscle target specific markers (epaxial, limb, and hypaxial) in proprioceptors that are present before proprioceptor subtype (Ia/II/Ib) development. The authors also identified the axon guidance molecules from the ephrin-A/EphA family as important regulators of proprioceptor guidance during development. In ephrin-A5 knockout mice, increased proprioceptor innervation was found in TA but not GS muscles. The identified epaxial, limb, and hypaxial specific markers should allow for future studies dissecting the role of proprioceptors innervating different muscle types. While more specific markers to different muscle groups will be helpful in those future studies, these identified markers are an excellent start.

Interestingly the authors also identified TrpV1 as a label for a subset of epaxial innervating proprioceptors. To my knowledge this is the first identification of TrpV1 in proprioceptors. In the discussion it would be helpful to discuss the potential significance of the presence of TrpV1 and whether the authors have any ideas about what role it might be playing.

Overall, this was a well done paper that identifies genetic markers that will be of use to other researchers interested in dissecting the role of proprioceptors from different muscle groups. The authors have used reasonable methods to confirm their putative markers. I can find no major flaws in the study and have only minor questions/suggestions.

Minor Comments:

On Fig 1 b & d it would be helpful to describe or show in the figure legend what, if anything, the different colors mean.

- We have used different colors just to simplify visual recognition of the cell clusters obtained after transcriptomic analysis. We clarified the significance of color coding in the figure legends.

Did you look at whether there were changes in epaxial innervation in the ephrin-A5 KO mice? In the discussion, you speculate that the mistargeted proprioceptors in the TA muscle may have been intended for epaxial muscles. If so, wouldn't you expect decreased innervation in epaxial muscles as well?

- We have done the suggested experiment and did not find any difference in the number of pSN labelled after epaxial muscle injection (ES) in *Efna5* $-/-$ mice (Extended Data Fig. 6h-k).

Reviewer #3:

*This manuscript by Dietrich et al. described molecular heterogeneity of pSNs at two developmental stages corresponding to E15.5 and P1, from thoracic and lumbar levels. For this, the authors are using scRNAseq with Cell-seq2 method and show that DRG cells from the above regions can be clustered into known cell types and then subclustered into proprioceptive neurons clusters, which they suggest might represent muscle-group specificity. They further attempt to correlate those subgroups of pSNs to axon guidance molecules, and tested Ephrin A5 (*Efna5*) expression as a requirement for pSN subgroups axon targeting. Beyond methodological considerations (see below), **a significant problem with this study is that it does not present a demonstration of any pSN subtype identities.***

- This is not true. We identified molecular signatures of proprioceptors innervating three cardinal "muscle" subtypes (back, abdominal, and hindlimb) and validated them at different developmental time points, using multiple experimental approaches at molecular and anatomical levels (more on it in the responses to specific criticisms). This is also clearly appreciated by the comments of Review #1 and #2.

Major concerns:

Omic:

While the study is presented as based on scRNAseq data, no

scRNAseq data analysis have been performed but one in fig3 e and is very limited (as it is, most of the genes cited throughout the manuscript seem to be selected by the authors but do not arise from RNAseq analysis per se).

- We selected the markers based on the list of top differentially expressed genes from analysis of the p1 scRNA-seq dataset (Fig. 3e). We would be happy to include and/or provide any other analysis that is requested by the Reviewer.

- data on gene expression per se are missing throughout the manuscript, including those related to the efficiency (gene coverage) of their method and to the number of cells included in the clustering (see below). Also, the average of gene expression per cell and per cluster is missing.

- We have included data on gene expression in Extended Data Fig.1 and Extended Data Fig.3. The number of cells included in the clustering was already available in the first version of manuscript submitted (Lines 99-101 177-183, and 473-476), in addition we have now included visual representations in Extended Data Fig.1 and Extended Data Fig.3.

- the manuscript contains only few heatmaps showing selected known markers and one heatmap of top expressed genes in fig3e.

- We provided the heatmaps that are necessary for understanding our work, we would be happy to provide more if needed to address any specific question. In the new version we added heatmap of differential gene expression analysis at e15.5 (Fig. 1g).

The expression unit is not specified.

- We have added expression units in figure legends.

- The actual number of pSNs clustered and used for further analysis, for E15.5 and for P1, is unknown.

- The number of cells included in the clustering was stated both in the results and methods sections (Lines 99-101 177-183, and 473-476).

These numbers seem very low, considering the panel 1d for instance.

- We sequenced 960 cells at e15.5 and 576 at p1. Considering the relative low abundance of proprioceptors in DRG (about 10%), especially for the epaxial subset targeted by the intersectional strategy (about 1%) and the difficulty to sort them while keeping the neurons healthy (for the p1 timepoint we had to resort to manual picking), these are reasonable numbers and in line with the one used but in similar work by Oliver et al., 2021, Nat Commun addressing the receptor identity of proprioceptors.

Importantly, clustering in Fig.1d seems arbitrary; no analysis is performed, no molecular markers (or top expressed genes or clusters 'exclusive genes expression) are shown, and no confirmation is provided in situ,

- Confirmation of the markers for “muscle” identity at e15.5 is provided in Figure 4d, e, f. Trpv1 expression in “back” proprioceptors is also extensively validated in Fig. 1, Extended Data Fig. 1, Fig. 2 and Extended Data Fig. 2.

yet the authors claim those are related to muscle specific groups and name them (i.e. fig3 and l181-187, the cluster assignment is arbitrary).

- The analysis in Figure 3 is related to the p1 dataset and not the e15.5 shown in “Fig.1d” as stated above by the Reviewer. Regardless, all the markers found in the p1 analysis are thoroughly validated using fluorescent in situ hybridization in three different mouse models: *Trpv1-Cre/Pv-Flp* intersectional line, *Pv-Cre* line, and in wild-type mice (Fig. 3f, g, h; Extended Data Fig. 3F; Extended Data Fig. 3i, j, k). In addition, we also validate the top candidates using smFISH analysis in proprioceptors retrogradely labelled from back and limb muscles at both e15.5 and p1 (Fig. 4; Fig. 5c, d).

- Why the authors change from E15.5 to p1 scRNAseq is unclear and p7 for anatomical analysis of the spinal cord and muscles.

- Because at p1 we can specifically sort proprioceptors connected to back muscles using the *Trpv1-Cre/Pv-Flp* intersectional mouse line, thus providing a unique tool for discovery and validation, as shown by the data. We use p7 for analysis of *Trpv1-Cre/Pv-Flp* intersectional line because the reporter expression in MS and GTO is brighter.

Inconsistencies:

1) it is unclear how the authors obtained so many dividing cells (obviously not neuronal) of unknown identity in their cell sorting-seq method in fig.1, if using PVcre;tomato to trace pSNs (PV is not expressed in progenitors).

- As clearly stated in our manuscript (lines 104-105) for this experiment we used the *Pv-tdTom* BAC mouse (Kaiser et al., 2016) and not *PVcre;tomato* as stated by the Reviewer. The dividing cells have signatures consistent with mechanoreceptors progenitor identity (*TrkB⁺*, *Maf⁺* and *Etv1⁺* and do not express glia markers; Lallemand and ernfors, 2012). Thus, the data indicate that using the *Pv-tdTom* BAC it is possible to label some progenitor neurons of the mechanoreceptive lineage that express *Pv* but have not yet exited the cell cycle.

2) It is not described how the authors came to focus on Trpv1, which, in contrast to what the authors write, seems in Fig. 1g to be enriched only in a part of pC6, of lumbar origin, and in a minority of pSNs in situ (Fig. 1i, only 5% at thoracic level).

- *Trpv1* is present in the differential gene expression analysis at e15.5 that we failed to present. We have included it the revised version (Fig. 1g) along with further *Trpv1* expression analysis in pSN clusters (Extended Data Fig. 1j). As explained in the text, *Trpv1* caught our eyes because it was not expected to be expressed in proprioceptors as it is a well-known *noc1*/thermoceptive neuron marker. We discussed this point in lines 128-131.

3) The use of many shortcuts and erroneous statements to claim generalities and support their conclusions.

• **For instance, the molecular identity of pSN subtypes has been demonstrated already, at embryonic, postnatal and adult stages (Oliver et al., 2021, Nat Commun; Wu et al., 2021, Nat Commun). Dietrich et al. however write that Ia, Ib and II subtypes are not existing prior to postnatal stages, citing Oliver et al. and Wu et al. as references. Wu et al. however clearly demonstrated Ia, Ib and II pSNs at E16.5. It is therefore unclear why Dietrich et al. claim the opposite.”**

- This is not correct. First, the manuscripts cited by Reviewer #3 focus on “receptor” identity and provides markers of Ia, Ib and II receptor types. Our work focuses on “muscle” identity. These are two different aspects of the final functional proprioceptive identity, as also appreciated by Reviewer #1 and #2 in their comments and stated throughout our manuscript. Second, both Oliver and Wu show that receptor identity is consolidated late during development, mostly postnatally. It is true that, Wu et al identify some markers for receptor types (Ia, Ib and II) at early stages (e16.5) but not for “muscle” types. Our transcriptomic analysis shows that “muscle signatures”, are already present at e15.5, thus preceding the emergence of even the early signs of receptor identity (e16.5). In agreement, Oliver et al presents analysis at e14.5 where clear signatures of receptor identity cannot be found. In addition, we performed gene expression correlation analysis for the Ia, Ib and II signatures identified by Oliver et al., and Wu et al., in our datasets and we do not find correlation at e15.5 and only modest correlation at p1 (Extended Data Fig. 5).

• **In line with the previous comment, the authors claim *Efna5* is not necessary for motor neurons muscle connectivity, citing Bonanomi et al. (2012, Cell), but Bonanomi et al. actually demonstrated in this study that it is essential for proper motor neuron muscle innervation, and sensory axon outgrowth depends on correct motor axon patterning (Wang et al., 2011, Neuron; Wang et al., 2014, Development).**

- This is not correct. Bonanomi et al., shows that elimination of both *Efna5* and *Efna2* is necessary to produce the motor neuron phenotype, while elimination of *Efna5* alone does not result in axon targeting defects (Bonanomi et al. 2012. Figures 1J-Q). Indeed, we confirmed this result in our own control experiment (Extended Data Fig. 6e-g) where we do not observe any motor neuron muscle innervation defects in *Efna5* knock-out mice. Thus, since there is no motor axon targeting defects after elimination of *Efna5*, this cannot be the cause of our proprioceptor phenotype.

Figure 1. Motor neuron targeting defects in *Efna2*; *Efna5* double mutants. The analysis of the allelic series shows that in order to have a motor neuron guidance defect elimination of both *Efna2* and *Efna5* is required. From Figure 1 of Bonanomi et al., *Cell* 2012.

Moreover, Wang et al., (2011) demonstrated that *Efna5* is expressed in epaxial sensory axons (the opposite in Dietrich et al.) at the time of axon outgrowth and is necessary for sensory axon innervation.

- Wang et al., (2011 Neuron) does not specifically show that *Efna5* is expressed in proprioceptive sensory axons or neurons. The authors report expression in whole DRG preparations at e11.5 (Figure S5F-G).

Figure 2. *Epha* and *Efna* expression in sensory and motor neurons at e11.5. The analysis shows expression by quantitative PCR of *Epha* and *Ephrin-A* in microdissected motor neurons and DRG. From Figure S5 of Wang et al., *Neuron* 2011.

In addition, the sensory phenotype reported is described by the authors in their own word as “sensory nerve loss” or “loss of epaxial sensory projections” is very different from the specific defect in proprioceptor muscle-target connectivity reported in our work. Finally, as in the case of the motor neuron guidance defect described by Bonanomi et al, the phenotype reported by Wang et al is evident only in *Efna2*; *Efna5* double mutants in a *Epha 3/4* double heterozygous background (Figure 5A-G).

Figure 3. Sensory neuron axon guidance defects in *Efna2*; *Efna5* double mutants. The analysis of the allelic series shows that in order to have a sensory neuron guidance defect elimination of both *Efna2* and *Efna5* (in an *Epha3/4* double heterozygous background) is required. From Figure S5 of Wang et al., *Neuron* 2011.

Dietrich et al. only analyzed expression of *Efna5* at a much later stage (P1, in *Trpv1* positive pSNs), which does not correspond to development of innervation (<E14).”

- We have transcriptomic analysis and validation of expression of *Efna5* in thoracic and lumbar proprioceptors at e15.5 which is shortly after proprioceptors innervate their specific muscle targets (Fig. 5a, b).

5) How the clusters found in Dietrich et al. relate to Ia, Ib and II pSNs is not investigated. Indeed, the authors extracted and sequenced all DRGs neurons expressing PV which for the proprioceptors includes all major pSN subtypes. In fact, the datasets of Oliver et al. and of Wu et al. (including at E16.5) are available online and can be used to understanding further specificity beyond Ia, Ib and II pSNs during early development.”

- This is a great point. We have analyzed correlation in gene expression for Ia, Ib and II markers identified by Oliver and colleagues as well as Wu et al., can colleagues in our datasets at e 15.5 and p1. In contrast to what observed for “muscle-type” signatures, the analysis shows that correlation in expression of “receptor-type” signatures is not present during embryonic development and only starts appearing at p1 (Extended Data Fig. 4).

6) Generalized markers to subtypes while their expression only covers part of cluster and groups of neurons; for instance, *Trpv1* is used for epaxial-pSNs while *Trpv1* is expressed only in 5% of thoracic pSNs.

- This is correct, but only in the case of *Trpv1*. In fact, we do not indicate *Trpv1* as general a marker for back proprioceptors (instead we propose and validate *Tox* and *Epha3*; Fig. 3, 4 and 5). *Trpv1* covers only a subset of back proprioceptors that we extensively characterized in the *Trpv1/Pv* intersectional line (Fig. 2 and Extended Data Fig. 2).

7) To analyze the molecular regulation of muscle-by-muscle or muscle groups innervation by pSNs, scRNAseq should be at least performed during the period of innervation. The phenotype they found in the *Efna5* null mice could be due to defect in motor axons, or earlier defects.

- As discussed in our response to Point #3, there are not defects in motor axons innervation in *Efna5* knock-out mice. Unfortunately, it is not possible to genetically label proprioceptors before e14.5 (Oliver et al., 2021) so we cannot conduct the analysis at earlier stages when proprioceptors innervate their muscle targets.

8) The mutant data points distribution for the number of MN labelled by injection of CTB in 2 different muscles GS and TA seem completely identical.

- The data points in Extended Data Fig. 5f (In the new version Extended Data Fig. 6f) are not “completely identical”. In addition, our result (absence of motor neuron muscle targeting defect in *Efna5* knock-out mice) recapitulate the findings of Bonanomi et al., Finally, we would be happy to provide original images and quantification.

9) CTB tracing present multiple advantages, one of those is that the positively traced cells are always very bright and easy to identify; in the panel fig5e, the fluorescence observed is uncommon with a lot of background (see lower panel), which is very similar to the fainted labelling in pointed cells.

- In general, I agree with the Reviewer, however it is also well-known that efficiency of filling can vary from experiment to experiment and depend on the identity of injected muscle (size and shape determine how much tracer can be injected safely without contaminating nearby muscles). In the images provided the difference between background and signal is clear. However, we would be happy to provide more examples, higher-resolution images, and raw data.

Other concerns:

- novelty wise, molecular markers such as *Tox*, *Chodl*, *Efna5* and *Gabrg1* were already known markers for pSN rostro-caudal distribution (Wu et al. 2021 Fig3).

- This is not completely correct. Wu et al, 2021 reported differential distribution of *Tox* and *Chodl* in DRG at different rostro-caudal levels (thus confirming our results) but not for *Efna5*, *C1ql2* or *Gabrg1*. In addition, the authors did not assign *Tox* and *Chodl* as specific markers of back and hindlimb muscle identities. This is discussed in lines 357-364.

- Moreover, a previous study in 2016 by Poliak et al. identified a series of muscle-type specific pSN markers using Affymetrix screening. It is surprising that the authors, here, did not attempt to compare these reference data.

- This is a good point. However, Poliak et al, only focused on the distal hindlimb compartment and identified a small set of markers that distinguish different muscles on the dorsoventral axis. We identified markers for major muscle compartments at thoracic (abdominal and back) and lumbar (back and hindlimb) levels.

With the revisions, additional experimental data, analysis and textual clarification, I hope that you will consider the paper for publication in Nature Communications.

Best regards,

Niccolò Zampieri

Reviewers' comments:

Reviewer #1 (Remarks to the Author):

The authors have done an excellent job responding to previous reviews by adding new figures, data and conducting new experiments to clarify past points that were unclear. The scientific quality of the manuscript was greatly enhanced from what I believe was already a high standard in the previous submission. It is unfortunate that the authors could not prove with new experiments their previous explanation that in ephrin 5 mutants back muscle proprioceptors were redirected to the limb. The authors have now changed the discussion to tone down explanations of this phenotype. This is appropriate but also leaves unresolved what is the exact origin of the excess TA proprioceptors in these mice. Despite this unanswered question I believe the amount of new data presented here will be very useful to future investigators and this will be an article that will be referred to in years to come because the cataloging of many interesting genes to label and study proprioceptors from specific muscle groups and/or different segments in the spinal cord.

One weakness is that the writing of this version was not as careful as the previous one and I identified a few mistakes, or I was confounded by some of the new figure references and the way they were explained. These are all minor points, but they need to be corrected or made clearer.

1. Line 128. Should extended data 1j-k be the correct figure reference here? 1g refers to C1-C5 data not pC1 to pC7.

2. Line 148. I would state in here the estimated percentage of VGLUT1 contacts that were genetically labeled in Trpv1; Pv ; tdT mice. From Extended data Fig. 2b, it looks something between 60 and 70%. This is a valuable piece of info. Please make sure the reader also knows these are VGLUT1 inputs to the cell body and ? may be? proximal dendrite?

3. Line 155 Extended data Fig. 2c. In the graph the Y-axis is labeled as back-pSNs to all pSNs, but this graph refers to genetic labeling of Trpv1 proprioceptors that is only a proportion of back pSNs. Therefore, the Y axis should be labeled more accurately, otherwise is highly misleading since it is unlikely that the proportion of back-pSNs to all proprioceptors is the same in lumbar and thoracic levels (as shown in following figures with Tox). Finally, I believe in the new text there is a distinction made between back (thoracic) and lower back (lumbar), so in essence there should be no back lumbar pSNs.

4. Line 156-157. I believe you also want to refer to Extended data Fig. 2d in here.

5. Figure 2D bottom graph. It will be useful to separate this graph into two, one for lumbar cells and the other for thoracic, so the difference is clearer.

6. Figure 3G and H. These figures do not match. In the lumbar region the data of lumbar 2 is excluded otherwise the distributions in G are impossible. I am Ok with "lu" representing L3 to L4, just make sure is the lower lumbar what is referred to and not the full lumbar region. If the authors have considered "lu" lumbar only L3 and L4 throughout all the analyses this should be made clearer in text describing the results

7. Lines 326 and 328. "Markers for back and abdominal subtypes at thoracic level (Tox and C1ql2) account for almost the entire proprioceptor population in thoracic DRG (~ 88%; Fig. 328 3g)," "thoracic" is redundant in this sentence, I think. Tox is not a marker of thoracic level because it is also in lumbar pC5 e15.5 cells (MMC at lumbar?).

Reviewer #2 (Remarks to the Author):

The authors have addressed all comments and questions to my satisfaction and I have no further questions.

Reviewer #3 (Remarks to the Author):

Remarks to the Author:

This manuscript, "The molecular foundation of proprioceptor muscle-type identity", uses transcriptomic and histological analysis to identify sub-types of proprioceptive sensory neurons in mouse dorsal root ganglia, and of mouse genetic tools to study how deletion of *Efna5* might affect muscle(-specific) sensory innervation. While the demonstration of few developmental states of proprioceptors is interesting, there are concerns regarding the limitation of the results.

Major concerns:

Sequencing, analysis, interpretation, and transparency.

The central question of this study is to reveal "molecular foundation of proprioceptor muscle-type identity".

For this, the authors decide to isolate and analyze only a couple hundred proprioceptive neurons (the exact number is unclear because not mentioned despite being asked for, and there is no description of the gene coverage, thus no information on the efficiency/quality nor on the average number of genes detected per cell...) for both facs sorted and picked cells. The sentence "cells pass the quality control" does not answer for the number of proprioceptors detected in the dataset. All information is required to be shared even more when asked for it. For details on how to present your data and perform analysis, I think that scRNAseq data-based stories published within the nature publish group for example might serve as strong examples.

Published literature demonstrates you need thousands of neurons with deep single cell sequencing, and even more neurons when using lower gene coverage methods. Without this, the data can only provide a small fraction of the actual genetic diversity of muscle type specific proprioceptors.

The main flaw when it comes to the results is that the number of cells together with the number of clusters is largely insufficient to make sense with the several dozens of muscle groups in one single limb and the only 4 clusters found in E16.5 lumbar data (of which certainly only two for the limb itself).

Is PV expressed in progenitor? Certainly not.

The tracing of PV lineage is questionable. The finding of 2 clusters with proliferative marks is unexpected, yet not clearly explained. The authors claim these are progenitors of mechanosensory cells however in the mouse line used, progenitors cells cannot be targeted as they never expressed PV. In the dorsal root ganglia (DRG) which contain the proprioceptors studied here, PV expression is restricted to proprioceptors population and is not expressed in the other neurons population nor glial cells forming the DRG. Moreover, PV expression onset is at E13.5 and onward.

For the authors knowledge with the aim that it would help comprehend the type of cell they have sorted and avoid wrong conclusion: 1/specific mechanosensory neurons progenitors do not exist, 2/ *MafA* and *TrkB* are neuronal postmitotic markers, and 3/ neurogenesis in the DRG is over for several days at E16.5 (for about 4-5 days for the mechanosensory neuron lineage), all sensory neurons have their peripheral innervation established.

Therefore, the authors cannot interpretate their cycling cell type as being the result of an efficient PV tracing. On the contrary, the detection of other cell type, obviously not neuronal because cycling, represents some degree of contamination.

Is *TrpV1* a marker of a subpopulation?

TrpV1 only marks 5% of proprioceptors at E16.5 in the authors dataset, and therefore is unlikely to define one of the presented clusters and specific group of muscle, i.e epaxial muscle proprio at

thoracic level should represent at least 50% of the population.

Difficulty comparing dataset with previously published dataset.

Published literature using single-cell transcriptomics of proprioceptive neurons (PV/RUNX3 and or Whirlin depending on stage) have sequenced thousands of proprioceptors and detected up to 10 000 genes per neurons (Oliver et al) and (Wu et al.), the difficulty in comparing dataset can unfortunately reflect low number of proprioceptors or low coverage, in which case the authors would benefit in drastically increasing the number of proprioceptors sequenced using recent sequencing platform. Novelty of the markers? Tox and Gabrg1 are known marker genes for proprioceptors. TrpV1 would need further investigations in particular in adult stage (see next paragraph).

Cellular state versus cellular subtypes?

Focusing a study on 2 developmental points (E16.5 and P1) where the animals cannot hold a posture or walk and interpret the data as subtype might be misleading. The developmental stage affects genes expression in different ways and focusing only on immature states might reflect temporary markers, developing and mature neurons are not comparable functionally and transcriptionally. Indeed, if the aim is to define muscle type specific proprioceptors, the authors cannot use developing neurons (E16.5 and P1) but should have analyzed the mature system instead (above one month old is best).

A large part of the molecular diversity of developing neurons is linked to the process of differential innervation and maturation, hence marks developmental state (not subtype), which they lose afterwards. While a much higher diversity (not shown here) is expected during development (because of the many different muscle types to innervate), this would be limited in adult; prior literature shows limited muscle type (or group) diversity in adult proprioceptors analyzing far more proprioceptors with very deep sequencing.

Is there a physiological reason to think of a higher diversity than the 7/8 proprioceptive neurons subtypes previously (recently) shown in adult?

Ephrins

While the observation of variable expression of ephrin family members is expected, the novelty of EfnA5 is not. In the Bonanomi paper, the authors choose at several occasion the double het mutant as control; and the panel Q that Dietrich et al. refer to clearly shows deficit in the EfnA5 mutant (with EfnA2 het). Regardless of prior work, in your manuscript Fig. 5F clearly shows a 2-fold increase in GS, yet not significant. This could simply be due to both its lower value and/or to the low number of animals analyzed?

Point-by-point response to Reviewer's comments

- Reviewer #1 states:

“The authors have done an excellent job responding to previous reviews by adding new figures, data and conducting new experiments to clarify past points that were unclear. **The scientific quality of the manuscript was greatly enhanced from what I believe was already a high standard in the previous submission.** It is unfortunate that the authors could not prove with new experiments their previous explanation that in ephrin 5 mutants back muscle proprioceptors were redirected to the limb. The authors have now changed the discussion to tone down explanations of this phenotype. This is appropriate but also leaves unresolved what is the exact origin of the excess TA proprioceptors in these mice. Despite this unanswered question **I believe the amount of new data presented here will be very useful to future investigators and this will be an article that will be refer to in years to come because the cataloging of many interesting genes to label and study proprioceptors from specific muscle groups and/or different segments in the spinal cord.**”

We thank the Reviewer for highlighting the “high standard” of the scientific quality of the manuscript as well as the impact of our work.

Minor points:

1) Line 128. Should extended data 1j-k be the correct figure reference here? 1g refers to C1-C5 data not pC1 to pC7.

We fixed it.

2) Line 148. I would state in here the estimated percentage of VGLUT1 contacts that were genetically labeled in Trpv1; Pv ; tdT mice. From Extended data Fig. 2b, it looks something between 60 and 70%. This is a valuable piece of info. Please make sure the reader also knows these are VGLUT1 inputs to the cell body and ? may be? proximal dendrite?

We added this information in the figure legend.

3) Line 155 Extended data Fig. 2c. In the graph the Y-axis is labeled as back-pSNs to all pSNs, but this graph refers to genetic labeling of Trpv1 proprioceptors that is only a proportion of back pSNs. Therefore, the Y axis should be labeled more accurately, otherwise is highly misleading since it is unlikely that the proportion of back-pSNs to all proprioceptors is the same in lumbar and thoracic levels (as shown in following figures with Tox). Finally, I believe in the new text there is a distinction made between back (thoracic) and lower back (lumbar), so in essence there should be no back lumbar pSNs.

We changed the labeling of the Y axis in Extended Data Figure 2c to “tdTom+ pSN/pSN”.

4) Line 156-157. I believe you also want to refer to Extended data Fig. 2d in here.

Yes, we changed the text accordingly.

5) Figure 2D bottom graph. It will be useful to separate this graph into two, one for lumbar cells and the other for thoracic, so the difference is clearer.

The graphs are separated by thoracic and lumbar segments.

6) Figure 3G and H. These figures do not match. In the lumbar region the data of lumbar 2 is excluded otherwise the distributions in G are impossible. I am Ok with “lu” representing L3 to L4, just make sure is the lower lumbar what is referred to and not the full lumbar region. If the authors have considered “lu” lumbar only L3 and L4 throughout all the analyses this should be made clearer in text describing the results.

Yes. We clarified in the figure legend that in Figure 3g we scored only neurons at L 3 and L4.

7) Lines 326 and 328. “Markers for back and abdominal subtypes at thoracic level (Tox and C1ql2) account for almost the entire proprioceptor population in thoracic

DRG (~ 88%; Fig. 328 3g), “thoracic” is redundant in this sentence, I think. Tox is not a marker of thoracic level because it is also in lumbar pC5 e15.5 cells (MMC at lumbar?).

Yes. We corrected the text accordingly.

- Reviewer #2 states:

The authors have addressed all comments and questions to my satisfaction and I have no further questions.

- Regarding Reviewer #3, I will first address his/her latest comments, provided after receiving the revised version of the manuscript on 02/09/2022:

1) “Most of my comments from the first submission still holds (please read early comments), this study is a bit limited.

“*This study is a bit limited*”. We think that the Reviewer did not fully understand the scope and the importance of the manuscript. The main finding of the manuscript is the discovery and validation of molecular signatures that define cardinal proprioceptor “muscle-type” identities, which despite several transcriptomic analysis recently published were still unknown, namely proprioceptors connected to back, abdominal and hindlimb muscles. We performed two scRNA-seq sequences, one at e15.5 that resulted in the surprising discovery of Trpv1 as a marker for a small subset of proprioceptors. Extensive anatomical validation shows that indeed transient Trpv1 expression during embryonic development defines a discrete subset of proprioceptors that is selectively connected to back muscle (Figure 2 and Extended Data Figure 2). This finding allowed us to design another screen, that was performed at p1 to take advantage of the ability to specifically sort this anatomically defined subset of Trpv1+ proprioceptors and compare it to all proprioceptors sorted using the general marker Parvalbumin in order to highlight molecular differences between proprioceptors connected to back, abdominal and hindlimb muscles. Indeed, we were able to identify and validate *in vivo* (using three different experimental approaches) molecular markers for proprioceptors selectively connected to back (*Tox*, *Epha3*), hindlimb (*Gabrg1*, *Efna5*) and abdominal muscles (*C1ql2*) (Figures 3, 4a, 4b, 5a-d, Extended Data Figure 3, Extended Data Figure 5a-d). These data are novel and a significant step forward compared to the previous transcriptomic efforts done on proprioceptors or somatosensory neurons in general (Wu et al., 2019; Sharma et al., 2020; Oliver et al., 2021; Wu et al., 2021)

In addition, we showed that in the e15.5 datasets these markers distinguish proprioceptors from thoracic and lumbar DRG (Figure 5a, 5b, Extended Data figure 1k, Extended Data figure 5a), validated *in vivo* back and hindlimb muscle connectivity for Tox+ and Gabrg1+

proprioceptors respectively (Figure 4e) and found that the “muscle-type” molecular signatures identified at p1 are already present at e15.5 (Figure 4c), preceding the emergence of “receptor-type” molecular signatures. Finally, we showed that ephrin-A5 signaling has a role in controlling proprioceptor muscle-target specificity, thus providing evidence that some of the molecules identified are not just markers but also effectors involved in determining a key aspect of muscle-type identity.

2) “Reading the new version of the manuscript I was happy to see that the authors provide info on cells number and gene coverage, but I could also see that the actual number of proprioceptors found and analyzed in figure 1 is low, 193 proprioceptors.”

We agree that compared to other studies with different scopes, such analysis of large, broadly defined, neuronal populations, such as entire areas of the nervous systems, using high throughput methods, the number of neurons analyzed in our work can be considered low. However, studies focusing on genetically and/or anatomically restricted subpopulations of neurons, analyze number of cells comparable to us. For example, Baek et al., Cell Reports 2019 analyses about 200 retrogradely labelled spinocerebellar neurons. Moreover, a very relevant example is readily available in the work on proprioceptor receptor-types from Oliver et al., 2021 Nat. Comm. Where a comparable “low” number of proprioceptors is analyzed. From the manuscript:

*“DRG were dissociated and single tdT+ neurons were purified through fluorescent activated cell sorting (FACS), followed by plate sequencing (Fig. 2a). **Neurons (480 in total) were sampled from three different experiments and derived from animals of either sex (totaling four males and two females). Cells with low gene detection (<2000 genes; 30 cells in total) or with significant contamination from attached satellite cells were eliminated by filtering for the satellite/Schwann cell markers Apoe and Mpz (cells with >10% of the Apoe/Mpz mean transcript level were removed from downstream analysis; 242 cells in total)** (Supplementary Fig. 4c).”*

and

“We also identified five minor clusters (comprising fewer than 15 cells), which we omitted from downstream analysis.”

Thus (without even accounting for the precise number of these last cells omitted from analysis) at most 208 neurons (480-30-242) were analyzed in this work. In addition, it is important to notice that the main findings obtained from this work were independently confirmed by another group (Wu et al., 2021 Nat. Comm.) that analyzed more proprioceptors (1109). Therefore, these data indicate that about 200 neurons are enough to find biologically relevant information.

3) Those neurons were assigned to 7 clusters but the authors still do not provide a validation for the existence of those 7 clusters in vivo.”

This is partially true, but as explained in point #1 our work focused on cardinal muscle identities (back, abdominal and hindlimb) that were found at p1. We biologically validated multiple markers for those identities at p1 but also at e15.5 (both by *in silico* analysis of the e15.5 dataset, Figure 5a, 5b, Extended Data figure 1k, Extended Data figure 5a, as well as *in vivo* by retrograde labeling experiments, Figure 4e). We agree that validation *in vivo* of each one of the seven clusters identified at e15.5 would be very interesting but goes beyond the scope of the current work.

4) The other limitation is that the study is based on developmental stages, so the limitation is the usefulness of the dataset, especially if that study is supposed to help decipher the contribution of muscle-specific proprioceptive feedback to motor control, the modulation of spinal cord sensorimotor circuits (...) the markers have to be expressed and still represent the same muscle group in the adult (or a stage when the animal walk and hold a posture).

We do not agree. As clearly demonstrated by our intersectional genetic approach (*Pv::Flp; Trpv1::Cre*) it is possible by using a marker identified as early as e15.5 to selectively label subsets of proprioceptors defined by their muscle connectivity. Thus, one could use the same genetic approaches to eliminate or modulate the function of these neurons *in vivo* during behavior and define their role in motor control. By extension, in the future, the introduction of Cre and Flp lines driven by the promoter of other genes we validated will open the way for the generation of a genetic toolbox for targeting proprioceptor subtypes according to their muscle connectivity and study their role in sensorimotor integration. In the words of Reviewer #1 from the first round of revision **“Most importantly the study generates key data sets for future genetic manipulations to probe the system”**.

5) The central novelty and interesting part of the paper is that Trpv1 is expressed in 5% of the back muscle innervating neurons at E15.5, (Trpv1 expression is not shown at P1). If its expression is not sustained in time, then even the relevance of the Trpv1 findings is limited.

This is not relevant for our study nor invalidate the main findings. We show that transient expression of Trpv1 (there is no expression at p1) at embryonic stages defines a subset of proprioceptors connected to back muscle allowing us to genetically target these cells, refine the transcriptomic analysis, and validate the findings *in vivo*. In addition, as discussed above, Trpv1

expression is used as a tool to genetically access a specific subset of proprioceptors provides proof of principle that genes in our datasets can be used to target proprioceptors according to their muscle connectivity.

- Next we will address the comments Reviewer #3 provided after reviewing for the second time, without realizing it, the original submission of the manuscript. From the decision letter sent on 26/082022:

- 6) Major concerns: Sequencing, analysis, interpretation, and transparency. The central question of this study is to reveal “molecular foundation of proprioceptor muscle-type identity”. For this, the authors decide to isolate and analyze only a couple hundred proprioceptive neurons (the exact number is unclear because not mentioned despite being asked for, and there is no description of the gene coverage, thus no information on the efficiency/quality nor on the average number of genes detected per cell...) for both facs sorted and picked cells. The sentence “cells pass the quality control” does not answer for the number of proprioceptors detected in the dataset. All information is required to be shared even more when asked for it.**

In the revised manuscript we did provide all the information that was asked for. Unfortunately, because of an editorial mistake, the original submission was sent out again and the Reviewer did not realize it. We already addressed in point #2 the concern about the number of cells analyzed

- 7) For details on how to present your data and perform analysis, I think that scRNAseq data-based stories published within the nature publish group for example might serve as strong examples. Published literature demonstrates you need thousands of neurons with deep single cell sequencing, and even more neurons when using lower gene coverage methods. Without this, the data can only provide a small fraction of the actual genetic diversity of muscle type specific proprioceptors.**

Oliver et al., 2021 and Wu et al., 2021 (both published in Nature Communications) came to the same conclusions by analyzing respectively 208 and 1109 proprioceptive neurons. Thus, obviously, there is not always the need for “*thousands of neurons*”. We do have high gene coverage (Extended Data Figure 1d, e; Extended Data Figure 3c, d).

- 8) The main flaw when it comes to the results is that the number of cells together with the number of clusters is largely insufficient to make sense with the several dozens of muscle groups in one single limb and the only 4 clusters found in E16.5 lumbar data (of which certainly only two for the limb itself).**

As detailed in point #1, we focused on the finding that at p1 we could discriminate between major muscle subdivisions (back, abdominal and limb) and validated those. We think it is interesting that at e15.5, when muscle connectivity has just been established, there is more variability (we found 7 cluster of which 4 of lumbar origin) which could hint at more refined aspects of muscle identity such as functional- (i.e. flexor vs. extensor) or anatomical-types (i.e.: proximal vs. distal muscle), indicating that indeed as suggested by the Reviewer by increasing the amount of neurons analyzed at early embryonic stages it may be possible in the future to get to markers at a single muscle level.

9) Is PV expressed in progenitor? Certainly not. The tracing of PV lineage is questionable. The finding of 2 clusters with proliferative marks is unexpected, yet not clearly explained. The authors claim these are progenitors of mechanosensory cells however in the mouse line used, progenitors cells cannot be targeted as they never expressed PV.

Our data do not show *Pv* expression in progenitor cells (Figure 1c). In addition, we only report that we found two clusters expressing proliferative markers “*while C2 and C4 are characterized by proliferation markers (Mki67+, Mcm2+, and Pcna+)*”. Most importantly, regardless of the nature of the cells in C2 and C4 these clusters were excluded from further analysis as we the paper focuses only neurons in C1 which are bona fide postmitotic proprioceptors expressing high levels of *Pv*, *Runx3*, *TrkC*, *Etv1*, *Avil* and *Isl1* (Figure 1C).

10) Is TrpV1 a marker of a subpopulation? TrpV1 only marks 5% of proprioceptors at E16.5 in the authors dataset, and therefore is unlikely to define one of the presented clusters and specific group of muscle, i.e epaxial muscle proprio at thoracic level should represent at least 50% of the population.

Yes it is and it is extensively proven by our lineage tracing experiments (Figure 2 and Extended Data Figure 2). However, as correctly stated by the Reviewer it is only a subset of back-innervating proprioceptors at thoracic level. We found labelled at p7 thoracic levels about 10% of all proprioceptors (Extended Data Figure 2c). Since at thoracic level about 50% of all are expected to be connected to back muscles, lineage tracing with the intersectional *Pv::Flp; Trpv1::Cre* lines cover about a 20% all back proprioceptors . We acknowledge this fact in the result section and would be happy to further clarify. Most importantly, this does not change our conclusion as in fact we do not include *Trpv1* in either the markers for back-innervating proprioceptors (Figures 4c, d) nor in their molecular signatures (Lines 324-326): “*We identified and validated molecular signatures for proprioceptor innervating cardinal muscle groups: back (Tox, EphA3), abdominal (C1ql2), and hindlimb (Gabrg1, Efna5).*”

11) Difficulty comparing dataset with previously published dataset. Published literature using single-cell transcriptomics of proprioceptive neurons (PV/RUNX3 and or Whirlin depending on stage) have sequenced thousands of proprioceptors and detected up to 10 000 genes per neurons (Oliver et al) and (Wu et al.), the difficulty in comparing dataset can unfortunately reflect low number of proprioceptors or low coverage, in which case the authors would benefit in drastically increasing the number of proprioceptors sequenced using recent sequencing platform.

This statement is simply not true and addressed earlier in points #2 and #7 (Oliver et al., 2021; analyzed 208 neurons and Wu et al 2021; 1109 neurons). We also have similar number of genes per neuron (Extended Data Figures 1d, e and Extended Data Figure 3c, d).

12) Cellular state versus cellular subtypes? Focusing a study on 2 developmental points (E16.5 and P1) where the animals cannot hold a posture or walk and interpret the data as subtype might be misleading. The developmental stage affects genes expression in different ways and focusing only on immature states might reflect temporary markers, developing and mature neurons are not comparable functionally and transcriptionally. Indeed, if the aim is to define muscle type specific proprioceptors, the authors cannot use developing neurons (E16.5 and P1) but should have analyzed the mature system instead (above one month old is best). A large part of the molecular diversity of developing neurons is linked to the process of differential innervation and maturation, hence marks developmental state (not subtype), which they lose afterwards. While a much higher diversity (not shown here) is expected during development (because of the many different muscle types to innervate), this would be limited in adult; prior literature shows limited muscle type (or group) diversity in adult proprioceptors analyzing far more proprioceptors with very deep sequencing. Is there a physiological reason to think of a higher diversity than the 7/8 proprioceptive neurons subtypes previously (recently) shown in adult?

Please see answer to point #8.

13) Ephrins While the observation of variable expression of ephrin family members is expected, the novelty of Efna5 is not. In the Bonanomi paper, the authors choose at several occasion the double het mutant as control; and the panel Q that Dietrich et al. refer to clearly shows deficit in the Efna5 mutant (with Efna2 het).

This is not true. The finding that proprioceptors connected to different muscles (Back and limb) express different ephrins signaling molecules (Epha3 and Efna5) is novel. Second, there is no motor neuron connectivity phenotype in the Efna5 ^{-/-} mice, we clearly show it in our experiments (Extended Data Figures 6h-k). This is in line with the findings described by Bonanomi et al., 2012 where phenotypes are observed only in allelic combinations where Efna2 is removed in combination with Efna5 but not in Efna5 ^{-/-} single ko mice.

14) Regardless of prior work, in your manuscript Fig. 5F clearly shows a 2-fold increase in GS, yet not significant. This could simply be due to both its lower value and/or to the low number of animals analyzed?

Yes, that is a possibility. We favor the hypothesis that the phenotype is not significant for GS because only about 60% of GS proprioceptors express Efna5 (Figure 5c, 5f and 5g). Probably due to the fact that the GS is composed by two distinct muscles, the medial and the lateral, however because they are just next to each other it is very difficult to specifically label only one and in our experiments we target both.

Reviewers' comments:

Reviewer #1 (Remarks to the Author):

The authors have satisfactorily addressed all my questions. My previous opinion about the high significance of this study for future work in the field remains the same as in previous commentaries.

Reviewer #4 (Remarks to the Author):

The main strength of this work is (a) the finding and the characterization of anatomically localized proprioceptive subtypes. While sensory neuron diversity at this level was assumed and has been sought by others, this is (as far as I am aware) its first description at a molecular/genetic level. This represents an exciting development and will open the door to many new future studies. Other strengths include (b) the finding that this diversity precedes receptor specialization, though this is expected given that axon guidance to proper peripheral and spinal targets is known to occur well beyond the postnatal stages of Type I/II diversification. (c) I expect that the field will be happy to use TrpV1 as a genetic access point for anatomically restricted (back muscles) proprioceptor neurons. And (d) as a bonus, Extended Data Figure 3f is gorgeous and very interesting.

A weakness of this work is the analysis of the sequencing data, which will somewhat limit the utility of the data as a resource that can be mined by others.

1. Data exclusion criteria: The low-quality filters seem overly stringent and atypical leading to elimination of approximately half of all cells (441/960 and 332/576). How were the 0.3 quantile criteria set? What was the distribution of nGene and nUMI before filtering? Since these parameters can scale with cell size or cell identity (indeed, this seems to be the case here from Extended Data Figure 1e), what measures were taken to ensure that the exclusion step did not bias the data?
2. How were clustering parameters selected (or tested) and how robust are the detected clusters? For example, it seems that pC2, pC4, and pC7 may not be truly distinct (or actually contain overlapping sub-populations). The relatively small dataset complicates the interpretation of these smaller, less studied clusters and a note about this should be included in the discussion.
3. What statistical test and parameters were used to find markers (line 476)?

Minor points:

4. It would be helpful to include "feature plots" for the main marker genes analyzed, similar to what is shown for Hoxc10 in the Extended Data.
5. The cluster nomenclature is a bit confusing. It would be helpful to assign distinct names to the clusters at different stages. For example, "C1" is used to identify a cluster from e15.5 and a cluster from p1 and "pC1" is also used as a label for a subtype of the e15.5 data.
6. In line 249-251, the authors describe Tox in the e15.5 dataset, but I only see Tox2 in Figure 1; while Tox is shown in Extended Data Figure 3i, it also appears to be in C4.
7. Were replicates performed for the cell picking and/or are these results pooled from multiple animals.

Point-by-point response to Reviewer's comments for manuscript NCOMMS-22-11523D

We thank the Reviewers for their comments highlighting the significance, strengths and remaining weaknesses of the work. We have addressed these concerns in order to further improve the manuscript.

Reviewer #1 (Remarks to the Author):

The authors have satisfactorily addressed all my questions. My previous opinion about the high significance of this study for future work in the field remains the same as in previous commentaries.

Reviewer #4 (Remarks to the Author):

The main strength of this work is (a) the finding and the characterization of anatomically localized proprioceptive subtypes. While sensory neuron diversity at this level was assumed and has been sought by others, this is (as far as I am aware) its first description at a molecular/genetic level. This represents an exciting development and will open the door to many new future studies. Other strengths include (b) the finding that this diversity precedes receptor specialization, though this is expected given that axon guidance to proper peripheral and spinal targets is known to occur well beyond the postnatal stages of Type I/II diversification. (c) I expect that the field will be happy to use TrpV1 as a genetic access point for anatomically restricted (back muscles) proprioceptor neurons. And (d) as a bonus, Extended Data Figure 3f is gorgeous and very interesting.

A weakness of this work is the analysis of the sequencing data, which will somewhat limit the utility of the data as a resource that can be mined by others.

- For our analysis we elected to keep only high-quality cells. We made raw data and processed data available to the community (NCBI GEO data repository database accession code GSE190605) so that this resource can be mined using different criteria or keeping the same described in the publication.

1. Data exclusion criteria: The low-quality filters seem overly stringent and atypical leading to elimination of approximately half of all cells (441/960 and 332/576). How were the 0.3 quantile criteria set?

- Different filter distributions (quantile 0.1 to 0.9) were used to identify cells of the best quality in both sequencing runs, and we then evaluated the markers and data distributions to ensure quality of the experiments while retaining enough information to draw conclusions. In this regard, a minimum of 7,722 total counts/cell and 2,045 genes/cell were set (p1 dataset: 576 cells), corresponding to the 0.3 quantile, and a similar threshold was used for the e15.5 experiment (960 cells, 5,947 gene/cell, and 22,444 total counts/cell). Despite reducing the number of cells, we found that this threshold ensured high data quality, reducing potential bias in the downstream analysis.

What was the distribution of nGene and nUMI before filtering?

- We observed wide nGene distributions ranging from 45 to 15,141 genes/cell in the e15.5 dataset (960 cells, avg.: 7,458 genes/cell) and 44 to 14,752 genes/cell in the p1 dataset (576 cells, avg.: 5,626 genes/cell). nUMI distributions were similarly wide ranging from 48 to 442,050 total counts/cell in the e15.5 dataset (960 cells, avg.: 60,906 total counts/cell) and from 75 to 370,449 total counts/cell in the p1 dataset (576 cells, avg.: 54,449 total counts/cell).

Since these parameters can scale with cell size or cell identity (indeed, this seems to be the case here from Extended Data Figure 1e), what measures were taken to ensure that the exclusion step did not bias the data?

- The Reviewer makes an interesting observation as indeed cluster 2, 3 and 4 at e15.5 that associated with different mechanoreceptor identity (*Maf+*, *Ntrk2+*) present a lower gene count compared to proprioceptors (e15.5 cluster 1 *Ntrk3+*; Supplementary Fig. 1e). Following the removal of low-count cells, the data was normalized using scran's `computeSumFactors` and scater's `logNormCounts` functions (with the default parameters) to avoid potential bias in the distribution of counts due to differences in coverage.

2. How were clustering parameters selected (or tested) and how robust are the detected clusters? For example, it seems that pC2, pC4, and pC7 may not be truly distinct (or actually contain overlapping sub-populations). The relatively small dataset complicates the interpretation of these smaller, less studied clusters and a note about this should be included in the discussion.

- To define the cell populations contained in the experiment, we specifically evaluated the clustering output by testing different thresholds. We commented on this point in the discussion (Line 321-323).

3. What statistical test and parameters were used to find markers (line 476)?

- The markers were obtained using `findMarkers` functions from scran package (`test.type="t", direction = up`). We defined the markers of a cell populations based on $FDR < 0.05$ and $\log_2FC > 1$.

Minor points:

4. It would be helpful to include "feature plots" for the main marker genes analyzed, similar to what is shown for *Hoxc10* in the Extended Data.

- We have included feature plots for *Tox*, *Epha3*, *Gabrg1*, *Efna5* and *C1ql2* (Supplementary Fig. 7).

5. The cluster nomenclature is a bit confusing. It would be helpful to assign distinct names to the clusters at different stages. For example, "C1" is used to identify a cluster from e15.5 and a cluster from p1 and "pC1" is also used as a label for a subtype of the e15.5 data.

- In order to avoid any confusion, we have assigned distinct names to different clusters at e15.5 and p1.

6. In line 249-251, the authors describe Tox in the e15.5 dataset, but I only see Tox2 in Figure 1; while Tox is shown in Extended Data Figure 3i, it also appears to be in C4.

- That is correct, we apologize for the confusion. Tox is not present in Fig. 1g as it does not satisfy the parameters chosen for the differential gene expression analysis shown there. We changed the text and do not refer to this figure any longer, but only to the violin plot in Supplementary Fig. 1. In addition, as noted by the reviewer we do also observe few cells in cluster C4 at p1 ("Ab-pSN") but at a lower level compared to cells in C2 and with parameter set for differential gene expression analysis does not represent a marker for cluster C4 (Supplementary Fig. 3i).

7. Were replicates performed for the cell picking and/or are these results pooled from multiple animals.

For scRNA-seq experiment at e15.5 cells from at least $n \geq 4$ embryos were pooled. Embryos from $n=6$ litters were used in total. For experiments at p1 cells were collected from individual animals. The following animal numbers were used for cell picking experiments at p1:

Trpv1Cre-Basbaum; PvFlp; Ai65:
thoracic DRGs (n=14 animals),
lumbar DRGs (n=15 animals)

Trpv1Cre-Hoon; PvFlp; Ai65:
thoracic DRGs (n=12 animals)
lumbar DRGs (n=12 animals)

PvCre; Ai14:
thoracic DRGs (n=5 animals)
lumbar DRGs (n=5 animals)

In summary, we have modified the text according to the comments of Reviewer #4 and added the data requested in Supplementary Fig. 7.

Best regards,

Niccolò Zampieri